

# ML4Fire-XGBv1.0: Improving North American wildfire prediction by integrating a machine-learning fire model in a land surface model

Ye Liu [1], Huilin Huang [1], Sing-Chun (Sally) Wang [1], Tao Zhang [2], Donghui Xu [1], and Yang Chen [3]

[1] Pacific Northwest National Laboratory, Richland, WA, USA.
[2] Brookhaven National Laboratory, Upton, NY, USA
[3] University of California, Irvine, California, USA

*Correspondence to*: ye.liu@pnnl.gov

## Abstract

Wildfires have shown increasing trends in both frequency and severity across the Contiguous United States (CONUS). However, process-based fire models have difficulties in accurately simulating the burned area over the CONUS due to a simplification of the physical process and cannot capture the interplay among fire, ignition, climate, and human activities. The deficiency of burned area simulation deteriorates the description of fire impact on energy balance, water budget, and carbon fluxes in the Earth System Models (ESMs). Alternatively, machine learning (ML) based fire models, which capture statistical relationships between the burned area and environmental factors, have shown promising burned area predictions and corresponding fire impact simulation. We develop a hybrid framework (ML4Fire-XGB) that integrates a pretrained eXtreme Gradient Boosting (XGBoost) wildfire model with the Energy Exascale Earth System Model (E3SM) land model (ELM) version 2.1. A Fortran-C-Python deep learning bridge is adapted to support online communication between ELM and the ML fire model. Specifically, the burned area predicted by the ML-based wildfire model is directly passed to ELM to adjust the carbon pool and vegetation dynamics after disturbance, which are then used as predictors in the ML-based fire model in the next time step. Evaluated against the historical burned area from Global Fire Emissions Database 5 from 2001-2020, the ML4Fire-XGB model outperforms process-based fire models in terms of spatial distribution and seasonal variations. Sensitivity analysis confirms that the ML4Fire-XGB well captures the responses of the burned area to rising temperatures. The ML4Fire-XGB model has proved to be a new tool for studying vegetation-fire interactions, and more importantly, enables seamless exploration of climate-fire feedback, working as an active component in E3SM.



## 1 Introduction

Recent wildfire outbreaks worldwide have raised alarms due to wildfires burning longer and more intensely in many regions, posing significant threats to human livelihoods and biodiversity. Over the globe, climate change has contributed to a 16% increase in the global burned area over the past two decades, while human influences, including ignition and suppression, have reduced by 27% (Burton et al., 2023; Jones et al., 2022). The continental United States (CONUS) has emerged as a hotspot for wildfires, where both climate change and human activities have fueled a 42% increase in the burned area (Jones et al., 2022). Such expansive burned areas release an average of 162 million tons of $CO_2$ and 0.9 million tons of $PM_{2.5}$ annually into the atmosphere, resulting in over \$200 billion health costs due to exposure to wildfire smoke (Samborska et al., 2024; JEC, 2023). Accurate prediction of wildfire risks has become an urgent need.

Traditional fire models, predominantly process-based models, simulate the behavior of individual wildfires using theoretical equations for ignitions and fire spread (Hantson et al., 2016). These models explicitly simulate the number and size of individual fires by incorporating parameterizations and parameters derived from laboratory or field experiments and typically estimate burned area by scaling up to the grid-cell level (Lasslop et al., 2014; Pfeiffer et al., 2013; Yue et al., 2014; Li et al., 2012; Thonicke et al., 2010; Huang et al., 2020, 2021; Arora and Boer, 2005; Burton et al., 2019). While process-based wildfire models have proven effective in simulating global burned area distribution (Hantson et al., 2020), they often fall short of accurately predicting the extent and temporal changes of wildfires over the CONUS (Forkel et al., 2019; Teckentrup et al., 2019). The climate and vegetation controls on the CONUS burned area and their relative importance are incorrectly represented, leading to failures in burned area predictions regarding both spatial distribution and temporal variations (Forkel et al., 2019). Human ignition and suppression are assumed to be linearly or log-linearly related to population density and the gross domestic product (GDP), respectively (Jones et al., 2023; Li et al., 2013). This assumption overlooks a more nuanced picture of human activities, such as road density, cultural differences, agricultural activities, and forest management policy (Jones et al., 2022; Villarreal et al., 2022; Hanan et al., 2021; Miller et al., 2009; Turco et al., 2023; Haas et al., 2022). Process-based fire models are often integrated with biogeochemical process-enabled land models (hereafter referred to as BGC model) within Earth system models (ESMs) to predict fire disturbances on carbon allocation, which is then used to update energy balance, water budget, and carbon fluxes in the land model. Incorrect simulation of burned areas over the CONUS induces large uncertainties in the assessment of fire impacts using ESMs.

Recent advances have explored the application of machine learning (ML) techniques in wildfire prediction (e.g., Wang et al., 2021; Buch et al., 2023; Li et al., 2023; Zhu et al., 2022). ML models offer the advantage of capturing nonlinear dependencies and complex interactions between driving factors and fire dynamics, without the need for explicit understanding of physical processes (Rodrigues and de la Riva, 2014). Zhu et al. (2022) presented a deep neural network (DNN) scheme that surrogated the process-based wildfire model with the Energy Exascale Earth System Model (E3SM) interface, demonstrating over 90% higher accuracy in simulating global burned area. Wang et al. (2021) combined the local predictors, large-scale meteorological patterns, and the eXtreme Gradient Boosting (XGBoost) algorithm to build an ML wildfire model, which





improves the temporal correlations of burned areas in several regions over the CONUS by 14–44%. Buch et al. (2023) developed a novel stochastic machine learning (SML) framework, SMLFire1.0, with a high spatial resolution of 12 km over the Western U.S. (WUS).

The newly developed ML fire models often focus on wildfire properties such as burned area, fire count, and fire emissions (Wang et al., 2021; Buch et al., 2023). Despite the improved fire predictions, fire impacts on the ecosystem, climate, and human community cannot be evaluated without integrating the wildfire process into the Earth system. In addition, climate change impacts on the burned area, either directly through fire weather conditions, or indirectly through ecosystem productivity, vegetation type, fuel loads, and fuel moisture – cannot be fully understood without explicitly representing the complex interplays between climate, ecosystems, and fire. For instance, a warmer and drier climate has been shown to cause an eightfold rise in the high-severity burned area from 1985 to 2017 over the WUS (Parks and Abatzoglou, 2020). The corresponding changes in fire dynamics may shift the vegetation species distribution from those originally low in resistance to wildfire to those in high resistance or even benefiting from regular fire occurrence (Rogers et al., 2015; Huang et al., 2024). The fire-adapted vegetation species, in turn, facilitate the frequent occurrence of wildfires. In this consideration, a full coupling of fire, ecosystem, and climate is required to better predict fire changes and the corresponding impacts in a future climate.

Leveraging the accuracy of ML-based wildfire models and the representation of ecosystem-climate interactions in ESMs, in this study, we have developed a novel hybrid framework to integrate a pretrained ML wildfire model with the E3SM land model (ELM) to study the full atmosphere-vegetation-wildfire feedbacks. This integration facilitates a dynamic feedback loop where outputs from the ML model (i.e., predicted burned areas) inform the land surface processes in ELM, which in turn update the inputs for the ML model for subsequent predictions. This approach leverages the detailed physical understanding of surface biogeophysical and biogeochemical processes provided by ELM and the predictive power of ML-based wildfire models to create a more accurate and robust framework for wildfire prediction and impact assessment. The remaining sections is arranged as follow: Section 2 introduces the ELM and ML wildfire model training method, coupling strategy, and datasets used in this study; Section 3 presents the simulated burned area compared with observations and several process-based fire models; discussion and conclusion are in Section 4.

## 2. Materials and methods

### 2.1 Model Description

#### 2.1.1 Default wildfire model in ELM

The ELM is part of the E3SM project which started with a version of the Community Earth System Model (CESM1). The ELM default wildfire module originated from the Community Land Model (CLM4.5) (Li et al., 2012). This wildfire model calculates burned areas by multiplying the number of wildfires and burned area per fire on a grid-cell level. The number of wildfires (fire count) is derived using anthropogenic and natural ignition sources, fuel load and combustibility, surface





meteorology, and anthropogenic suppression. The natural ignition source is derived from the number of cloud-to-ground
lightning flashes multiplied by a constant ignition efficiency (Prentice and Mackerras, 1977). Anthropogenic ignitions are
simply parametrized using a fixed number of potential anthropogenic ignitions by a person and population density (Venevsky
et al., 2002). Humans also suppress wildfires. The capability of fire suppression is assumed to be a function of gross domestic
product. The ignition efficiency is also altered by fuel conditions, including the fuel load (aboveground biomass) and fuel
combustibility (approximated using relative humidity, temperature, and top or root zone soil moisture). The spread of each fire
is approximated using an ellipse shape with its length-to-breadth ratio determined by wind speed and fuel moisture (Rothermel,
1972). This simple concept well captures the major constraints for predicting the global wildfire distribution and seasonal
variations (Rabin et al., 2017; Li et al., 2014; Huang et al., 2020).
Like many other process-based wildfire models, the default fire model in ELM benefits from the full ecosystem interactions
from its hosting land model, as well as the potential to be coupled with atmospheric models. With the BGC processes being
turned on, ELM-BGC reallocates carbon and nitrogen in leaf, wood, root, litter, and soil pools after fire based on plant
functional type (PFT)-dependent carbon combustion and mortality rate. The biogeochemical changes subsequently influence
biogeophysical properties such as leaf area index (LAI), vegetation canopy height, and albedo, disturbing the land-atmosphere
exchanges of energy and water fluxes. The post-fire vegetation recovery depends on the plant photosynthesis processes and
PFT competition strategy. The interactions between wildfire and vegetation under historical climate have been thoroughly
assessed in CLM long-term simulations (Li and Lawrence, 2017; Seo and Kim, 2023). The model framework is illustrated in
Figure 1. Hereafter the ELM coupled with the process-based fire model is referred to as ELM-BGC.

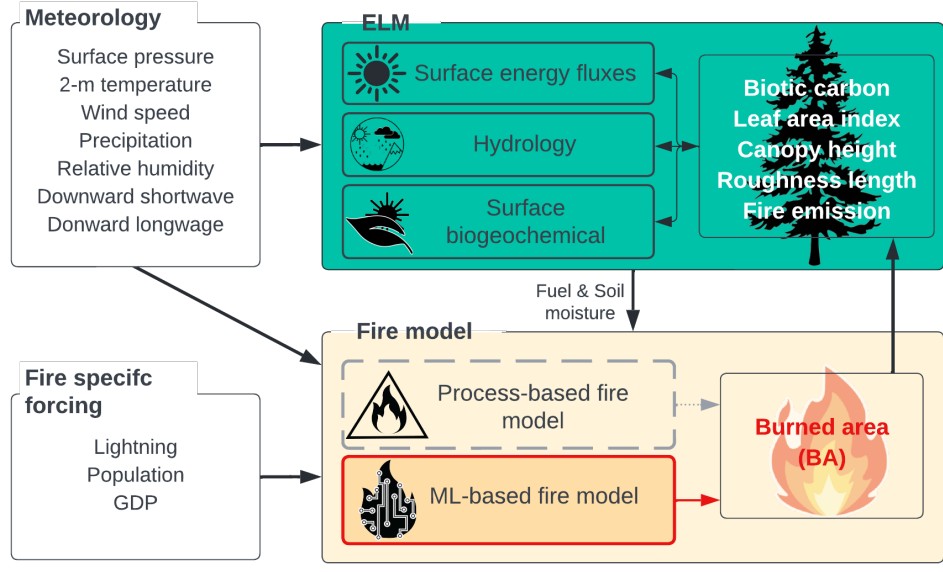


**Figure 1: Schematic diagram of the hybrid model framework.**



### 2.1.2 Machine learning wildfire model

The XGBoost-based wildfire model has proven to outperform process-based models in predicting burned areas over the CONUS (Wang et al., 2021). In this study, we tailored the pretrained XGBoost wildfire model to use variables directly provided by ELM at each grid cell. One modification is that we exclude the large-scale patterns used in Wang et al. (2021), without significantly affecting the model accuracy. XGBoost is a highly efficient and scalable implementation of gradient boosting, designed for performance and speed (Chen and Guestrin, 2016). It builds sequential decision trees to correct errors from previous models, using techniques like regularization to prevent overfitting and parallel processing for faster computation.

To reduce overfitting, we build a separate ML model for each year from 2001 to 2020 using the remaining 19 years' data. Model performance was evaluated based on its accuracy in predicting the spatial distribution and temporal variation of burned areas. Validation metrics included root mean square error (RMSE), mean absolute error (MAE), and the coefficient of determination ($R^2$). This pretrained XGBoost model is referred to as offline-XGB in the following analysis.

### 2.1.3 Hybrid modeling framework

The pretrained ML-based wildfire model is integrated with the ELM using the ML4ESM coupling framework. The ML4ESM framework offers a robust and flexible solution for integrating ML parameterizations into ESMs through a Fortran-Python interface (Zhang et al., 2024). It supports popular ML libraries such as PyTorch, TensorFlow, and Scikit-learn, enabling the seamless incorporation of ML algorithms to represent complex climate processes like convection and wildfire dynamics. The interface leverages C language as an intermediary for efficient data transfer by accessing the same memory reference, instead of the extra data copy or through files, minimizing memory overhead and computational inefficiencies. A C-Hub is then used to communicate variables from the Fortran-written ELM and the Python-written ML wildfire model. In our application, all surface meteorology, lightning, and socioeconomic data, alongside the ELM simulated fuel conditions are passed to the pretrained ML-based wildfire model to predict the burned area. The burned area is returned to ELM to calculate fire impacts and update surface properties.

## 2.2 Datasets and processing

### 2.2.1 Burned area datasets

The primary dataset for training and validating the ML-based model is the Global Fire Emissions Database version 5 (GFED5) (Chen et al., 2023). The GFED5 is a succession of GFED4s, by fusing multiple streams of remote sensing data to create a 24-year (1997-2020) dataset of the monthly burned area at 0.25° spatial resolution. During 2001-2020, the GFED5 comprises the Moderate Resolution Imaging Spectroradiometer (MODIS) MCD64A1 burned area product (Hall et al., 2016; Giglio et al., 2016, 2018), with adjustment for the errors of commission and omission. Adjustment factors are estimated based on region, land cover, and tree cover fraction, using spatiotemporally aligned burned areas from Landsat or Sentinel-2 (Claverie et al., 2018). Focusing on the MODIS-era, 2001-2020, we mask the grid cells with less than 30 months of non-zero burned



area (~two thirds of the total number of grid cells). This step is significant to avoid feeding the ML model with distinct predictor combinations while all zero burned areas.

Besides observations, we also obtained burned area from four state-of-the-art process-based wildfire models participating the Fire Model Intercomparison Project (FireMIP) (Rabin et al., 2017), including the Canadian Land Surface Scheme Including Biogeochemical Cycles (CLASSIC) (Melton et al., 2020), the Simplified Simple Biosphere model coupled with the Top-down Representation of Interactive Foliage and Flora Including Dynamics model (SSiB4-TRIFFID) (Huang et al., 2020, 2021), the SPread and InTensity of FIRE (SPITFIRE) coupled with the Organizing Carbon and Hydrology In Dynamic Ecosystems (ORCHIDEE) (Yue et al., 2014), and the Vegetation Integrative Simulator for Trace gases (VISIT) (Ito, 2019). The burned area simulation from the process-based fire model over the CONUS will be used to benchmark that from the hybrid framework.

### 2.2.2 Surface meteorological, lightning, and socioeconomic datasets

Surface meteorological variables including temperature, humidity, wind speed, downward shortwave radiation, downward longwave radiation, precipitation, and surface pressure are obtained from NLDAS-2 (Phase 2 of the North American Land Data Assimilation System) forcing fields to both drive the ELM and construct the training set for the ML fire model. This dataset combines multiple sources of observations (such as precipitation gauge data, satellite data, and radar precipitation measurements) to produce estimates of climatological properties at or near the Earth's surface at hourly temporal resolution and 1/8th-degree grid spacing. We use the temperature, relative humidity, specific humidity, wind speed, and precipitation directly from NLDAS-2 train the ML fire model. Additionally, we calculate the Standardized Precipitation Evapotranspiration Index (SPEI) following Beguería et al (2013) and vapor pressure deficit (VPD) based on NLDAS-2 dataset as additional input for ML model (Table 1). We coarsen this dataset to 0.25º to align with burned area datasets.

In addition to surface meteorological forcing, we acquire lightning and socioeconomic datasets from multiple sources. The 2-hourly climatology lightning flashes data from NASA LIS/OTD v2.2 at 2.5º resolution are used to calculate the number of natural ignitions. The gridded population density data is acquired from Kummu et al. (2018) while the GDP per capita is from the World Bank (https://data.worldbank.org/). All the datasets are resampled to 0.25×0.25 spatial and annual temporal resolution. To train the ML model, additional inputs including top-layer soil moisture, LAI, and spatial fraction of each plant functional types (PFTs) are simulated by ELM (explained further in Section 2.3).

**Table 1 Meteorological forcing, land surface properties, and fire specific inputs for training the XGBoost-ELM**

| Meteorological forcing | | Land surface property | |
|---|---|---|---|
| Temperature | NLDAS-2 | Soil moisture | ELM-BGC output |
| Relative humidity | | Leaf area index | |
| Wind speed | | Plant functional type (PFT) fraction | |
| Precipitation | | Fire specific inputs | |





| Standardized precipitation evapotranspiration index (SPEI) | | Lightning | NASA LIS/OTD v2.2 |
|---|---|---|---|
| Vapor pressure deficit (VPD) | | GDP | Kummu et al., (2018) |
| | | Population density | GPW v4 |

### 2.2.3 Ecoregion

We evaluate the model simulation of the burned area for each ecoregion adopted from the U.S. Environmental Protection Agency (EPA). Ecoregions are areas where ecosystems (and the type, quality, and quantity of environmental resources) are generally similar (Omernik and Griffith, 2014) and generally, wildfire properties in each ecoregion are similar. A combination of level I and level II ecoregions is used and some types have been combined to focus on the broad vegetation distribution. As shown in Figure 1, the Western Forested Mountains include NW Forested Mountains, Marine West Coast Forests, and Mediterranean California from ecoregion level 1. The North American (NA) Deserts include NA Deserts and small portions of Temperate Sierras and Southern Semi-Arid Highlands. The Northeast (NE) Temperate Forests include Mixed Wood Shield, Mixed Wood Plains, Central U.S. Plains, and Atlantic Highlands from ecoregion level II. The Southeast (SE) Temperate Forests include Southeastern U.S. Plains Ozark, Ouachita-Appalachian Forests, and Mississippi Alluvial and Southeast U.S. Coastal Plains ecoregion level II.

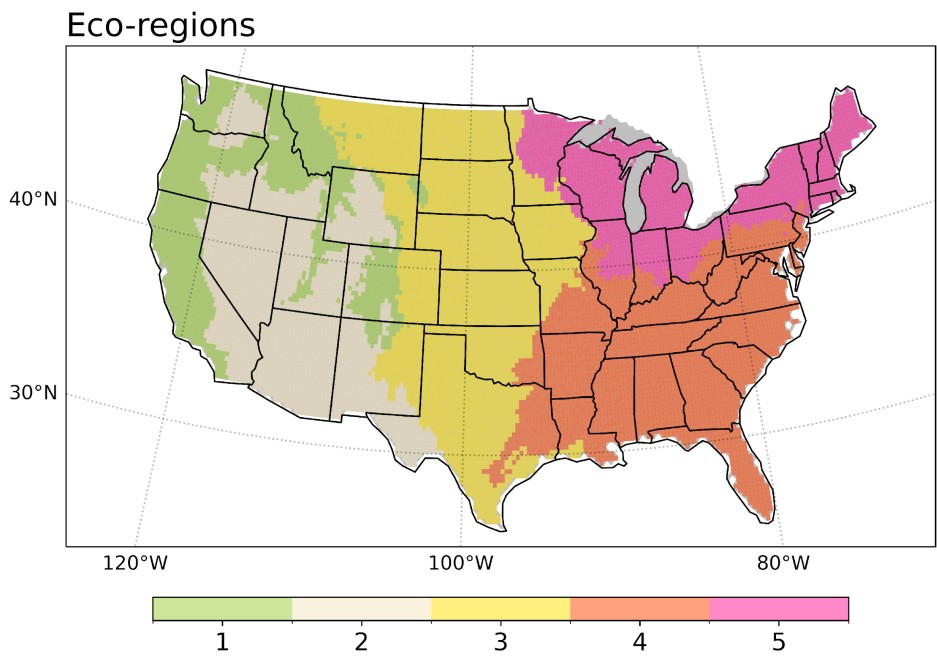

**Figure 2: Ecoregions used in fire model evaluation. 1 – Western Forested Mountains, 2 – NA Desert, 3 – Great Plains, 4 – SE Temperate Forests, and 5 – NE Temperate Forests.**



### 2.3 Model configuration and ML4Fire-XGB training processes

In ELM-BGC, vegetation properties, including canopy height and LAI, vary with carbon allocation and distribution, driven by climate variability and disturbances such as wildfires. To bring the model's carbon and nitrogen pools into equilibrium, we first conduct long-term spin-up simulations as suggested by Lawrence et al. (2011). We adopt a two-step approach consisting of a 400-year accelerated decomposition (AD) spin-up followed by a 400-year regular spin-up, driven by cycling NLDAS-2 meteorological forcing from 1981 through 2000. In the AD spin-up, acceleration factors will be applied to accelerate decomposition in soil organic matter pools, and for plant dead stem and coarse root mortality. The terrestrial carbon pools and vegetation distribution after spin-up simulations reach quasi-equilibrium states after the 800-year simulations.

Initialized with the quasi-equilibrium state from the spinup simulation, we conduct transient simulations with the process-based fire model in the ELM-BGC utilizing NLDAS-2 meteorological forcings for the period of 2001-2020. The surface soil moisture, LAI, fraction of each PFT output from ELM-BGC transient run are then used to train the offline-XGB prior to the coupled run within ELM. Furthermore, we run the coupled ML4Fire-XGB in which the pre-trained XGB model provides monthly burned area to ELM to update the land surface properties (LAI, PFT fraction, and soil moisture), which are then used as predictors in the ML-based fire model in the next time step. The differences in land surface properties input in offline-XGB and ML4Fire-XGB produce different burned area simulation, and the divergence accumulates over the 20-year simulation period.

In addition to the default transient simulations with ELM-BGC and ML4Fire-XGB which represent historical burned area, we conduct additionally sensitivity simulations with ELM-BGC and ML4Fire-XBG, utilizing the same NLDAS-2 meteorological forcings except for detrended temperatures to evaluate the responses of the modeled burned area to raising temperatures, which are considered as the primary driver of the increasing burned area over the WUS (Parks and Abatzoglou, 2020; Zhuang et al., 2021).

## 3 Results

### 3.1 Evaluation of the burned area spatial distribution

The spatial distribution of burned areas across the CONUS varies significantly (Fig. 3a), primarily influenced by climate, vegetation, and human activities. According to the GFED5, the CONUS experiences an averaged burned area fraction (BAF) of 0.6–0.9% yr$^{-1}$ (4.8–7.1 Mha yr$^{-1}$), which is consistent with Chen et al., (2023). High-burned areas are predominantly observed in the WUS (Western Forested Mountains and NA Desert), with BAF ranging between 0.4–0.9% yr$^{-1}$ (1.1–2.3 Mha yr$^{-1}$). States like California, Oregon, and Nevada, as well as the Rocky Mountain region including parts of Colorado and Wyoming experience large wildfires. The wildfires in the Pacific Northwest and northern California are generally lightning-caused and occur in boreal forests (Balch et al., 2017) whereas those in southern California are primarily caused by human ignition in dry forests and shrublands. The Southwest, including Arizona and New Mexico, also sees significant burned areas

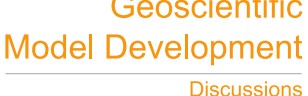

411 in shrublands and dry forests. In the Great Plains, states such as Kansa and North Dakota exhibit high burned areas, alongside

412 with Texas and Oklahoma, with a BAF ranging between 0.7–1.3% yr$^{-1}$ (1.6–2.9 Mha yr$^{-1}$). These high burned areas are

413 primarily contributed by agricultural fires, particularly for cleaning crop residues and managing pastures (Donovan et al.,

414 2020). The Southeastern U.S. experiences 0.9–1.5% yr$^{-1}$ (1.5–2.6 Mha yr$^{-1}$) BAF annually, while the temperate forested areas

415 covering Florida, Georgia, and the Carolinas, show lower burned areas compared to the West. This region often uses prescribed

416 burning to manage forests and reduce wildfire risk. The Midwest and Northeast exhibit sparse burned areas, with BAF mostly

417 less than 0.16–0.25% yr$^{-1}$ (0.2–0.3 Mha yr$^{-1}$).

418

**Figure 3: Observed and simulated burned area fraction (% yr$^{-1}$) averaged over 2001-2020. The dataset names are listed on the top of each panel.**

421 The offline-XGB wildfire model reproduces the burned area distribution over the CONUS well (Fig. 3b), with a spatial

422 correlation coefficient ($R_p$) of 0.96 ($p<0.01$) and a small bias (-0.4 Mha yr$^{-1}$). While integrated with ELM, the performance

423 holds ($R_p$=0.70, $p<0.01$, bias=1.0 Mha yr$^{-1}$) (Fig. 3d). This degradation is likely due to the vegetation-wildfire feedbacks. The



aboveground biomass and fuel moisture from ELM-BGC have been used to train the ML4Fire-XGB prior to the coupled run
within ELM. In the coupled simulation, ML4Fire-XGB updates the biotic carbon and fuel moisture based on the burned area
simulated in the previous timestep. Consequently, differences in the simulated burned area compared to the process-based
models are reflected in the biotic carbon and fuel moisture, accumulating over the 20-year simulation period and influencing
the burned area simulation in subsequent timesteps.
In various eco-regions, the offline-XGB model demonstrates minimal biases, and the ML4Fire-XGB model consistently
outperforms all process-based fire models in predicting annual mean burned area (Fig. 3b-f). The accurate simulation of burned
area over the Western Forest Mountains indicates that the ML4Fire-XGB framework generally captures the complex interplays
between climate, vegetation, and human activities, with both climate forcings and predicted vegetation status from ELM-BGC.
Meanwhile, the ML4Fire-XGB shows superior performance over the Great Plains, indicating that the ML model effectively
describes crop fire thereby utilizing data on crop fraction and LAI.
The performance of the five process-based fire models (ELM-BGC, CLASSIC, ORCHIDEE, SSiB4/TRIFFID, and VISIT)
in simulating burned areas over the CONUS shows both strengths and weaknesses (Figs. 3e-i and Fig. 4). All models generally
capture the high burned areas in key regions such as the WUS and Southeast U.S., except for ORCHIDEE which shows a
concentrated burned area in the Great Plains. However, these models tend to overestimate burned areas in regions across the
CONUS. ELM-BGC and SSiB4-TRIFFID-Fire have moderate overestimations over the CONUS, with 8.5 Mha yr$^{-1}$ and 11.1
Mha yr$^{-1}$, respectively. The burned areas are doubled in CLASSIC, ORCHIDEE, and VISIT simulations, with values up to
17.6 Mha yr$^{-1}$ (Fig. 4a).
In the Western Forest Mountains, where fuel is abundant due to dense forest coverage, all process-based models except
ORCHIDEE simulate 2 to 4 times of GFED5 burned area. This overestimation can be related to many factors including
overestimation of fuel combustibility and underrepresentation of anthropogenic fire suppression (Balch et al., 2017). In contrast,
wildfires in the NA Desert are primarily constrained by the fuel load. ELM-BGC and CLASSIC produce smaller
overestimations, while SSiB4-TRIFFID-Fire and VISIT significantly overestimate the burned area (4–15 times of GFED5),
likely due to overestimations of fuel load, which might be attributed to insufficient water stress on vegetation growth in the
arid region (Liu and Xue, 2020; Zhang et al., 2015). Although none of the process-based models accurately capture the spatial
distribution of burned area over the Great Plains (Fig. 1), ELM-BGC, SSiB4-TRIFFID-Fire, and VISIT produce comparable
burned areas to observations while CLASSIC and ORCHIDEE overpredict them (4–7 times of GFED5). The inaccurate
description of the spatial pattern and large intermodal spread in the Great Plains may be caused by inaccurate treatments of
cropland fires and pasture fires (Donovan et al., 2020). Notably, none of the process-based models has activated the explicit
cropland fire model. That says all vegetation models treat pastures as natural grasslands. This may explain the significant
overestimation of burned areas in ORCHIDEE as the SPITFIRE fire module has a much higher flammability in natural
grasslands compared to woody plants (Teckentrup et al., 2019). Therefore, information on how fuel properties and fire ignitions
differ between pastures and natural grasslands could help to improve burned area simulation in the process-based fire models
(Rabin et al., 2017). In the eastern U.S. (EUS) forests (Southeast and Northeast Temperate Forests ecoregions), fires are more



managed by prescribed burning, leading to fewer uncontrolled extreme wildfires. Consequently, all process-based models
perform reasonably well in these areas.

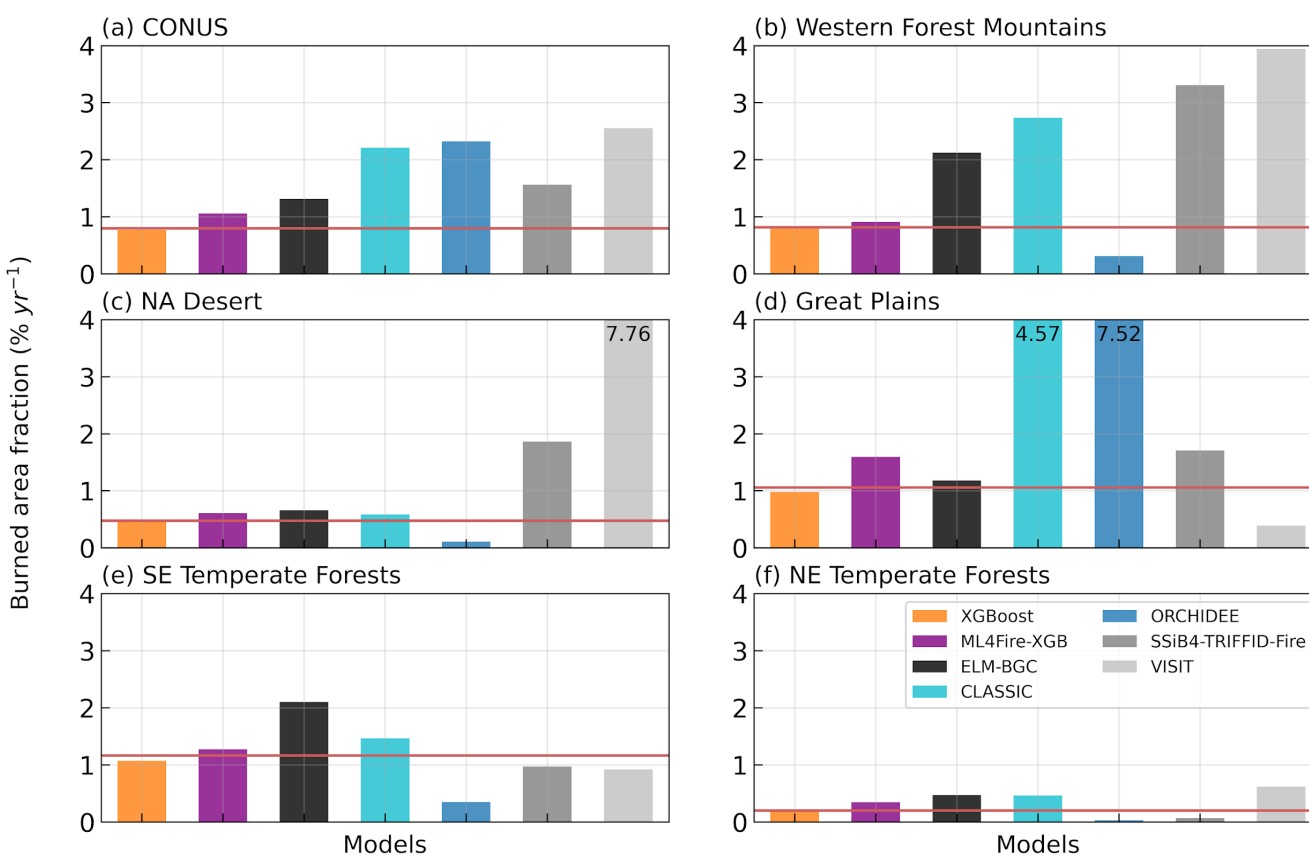


**Figure 4: Observed and simulated mean burned area fraction (% yr⁻¹) over the CONUS and eco-regions. The red line in each panel**
**indicates the observed burned area. Modeled burned areas greater than 4 % yr⁻¹ are truncated with the value denoted on the bar.**
### 3.2 Evaluation of the burned area temporal variability
We evaluate the model performance in simulating the monthly burned area and depicting fire seasons. Fire season is defined
as a monthly burned area greater than 1/12 of the annual total burned area. The COUNS has two fire seasons, i.e., March-
April-May and August-September-October, affected by both climate and human activities (Fig. 5a). The WUS fire season
spans from early summer to late fall, primarily determined by the dry conditions and high temperature during these months
(Safford et al., 2022; Schoennagel et al., 2017). Specifically, over the Western Forest Mountains, the fire season includes July
to November (Fig. 5b). Most models capture the July to October fire season, except for ORCHIDEE (May-August). However,
only offline-XGB, SSiB4-TRIFFID-Fire, and CLASSIC simulate the peak fire month in August, while others simulate a peak
~ 1–2 months late. Similar fire season and model performance are observed over the NA Desert (Fig. 5c). In wildfire-dominant



regions, the shift in fire peak months might be related to the representation of seasonality in vegetation production and fuel build-up in the BGC model (Hantson et al., 2020).

Human activities can also change the timing of fire occurrence (Le Page et al., 2010). Over the Great Plains, pasture fires are conducted during late winter to early spring to control pests, recycle nutrients, and prepare fields for planting (Gates et al., 2017). During the late summer to early fall, crop fires are conducted to clear crop residues. However, sometimes these fires can become uncontrolled, leading to larger wildfires that significantly impact the region. The fire seasons due to pasture fires and crop fires are evident in observations and are captured in offline-XGB and ML4Fire-XGB, despite ML4Fire-XGB slightly underestimating the peak in March. None of the process-based models is able to simulate these periods, instead, a summer fire season is predicted. In SE Temperate Forests, routinely prescribed burns reduce large fire occurrences across the year (Mitchell et al., 2014). The dry condition and/or fallen vegetation fuel larger burned areas in February–March and September–November. The ML-based models generally reproduce the fire seasons in March–April and September–November while none of the process-based models captures the bimodal seasonality. The results of NE Temperate Forests are similar to Great Plains, expect no peak burned area appears in November. The offline-XGB and SSiB4-TRIFFID-Fire capture the spring peak. Our evaluation suggests that the inclusion of anthropogenic fires could help to improve model simulations in Central and Eastern US. However, this requires a better understanding of how fire is used for land management under different socioeconomic and cultural conditions (Pfeiffer et al., 2013; Li et al., 2013).

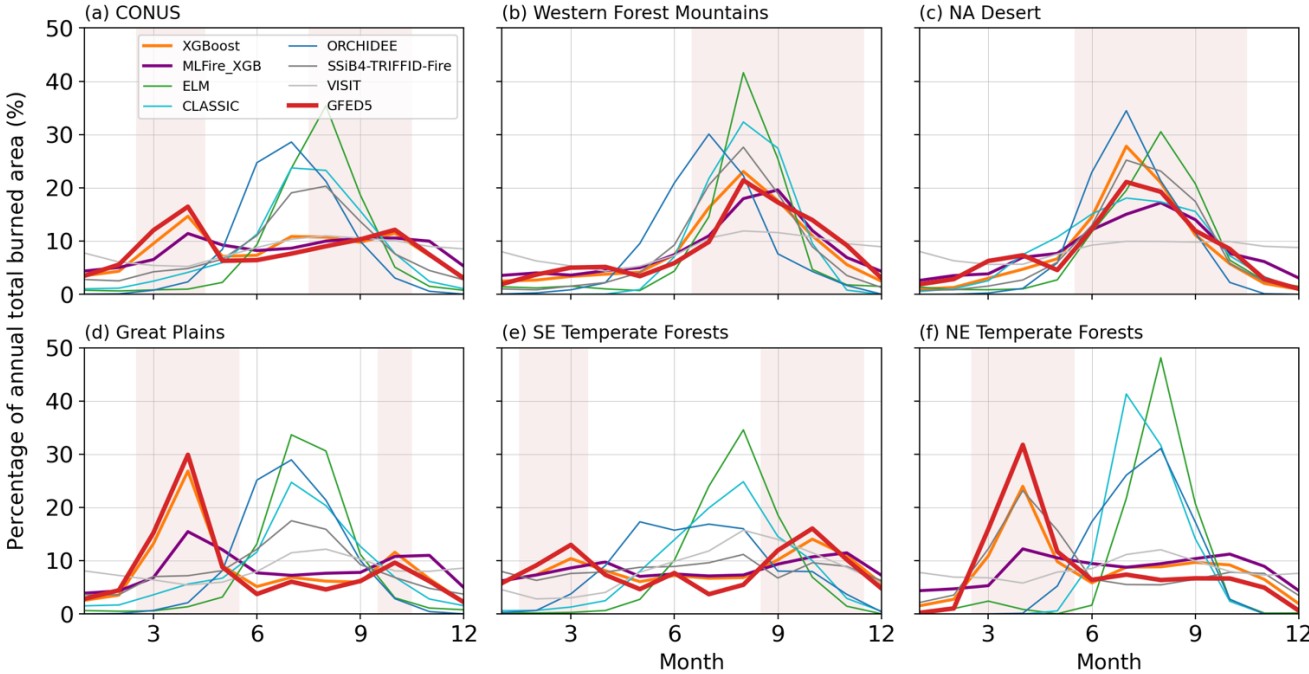

**Figure 5: Monthly mean burned area fraction (% yr⁻¹) over each eco-region.**



Over the CONUS, the observed interannual variability (IAV), measured using standard deviation, is 0.7 Mha yr$^{-1}$, accounting for 12% of the annual total burned area (Fig. 6a). Process-based models largely overestimate the IAV, ranging in 2.6–6.6 Mha yr$^{-1}$. This overestimation can be partly attributed to the overestimation in the annual total burned area. The relative IAV regarding the modeled annual mean value, ranging from 21% to 50%, is still overestimated by the process-based models. The machine learning models, offline-XGB and ML4Fire-XGB produce IVA of 0.9 Mha yr$^{-1}$ (15%) and 1.0 Mha year$^{-1}$ (12%), respectively.

Despite the magnitude of IAV being amplified by process-based models, after extracting the mean values and dividing by standard deviation, the standardized time series well correlated with the observation (Fig. 6b). Since the modeled IVA is generally influenced by climate variability and the climate-driven fuel variability, both process-based and ML-based models capture the timing of the fluctuations.

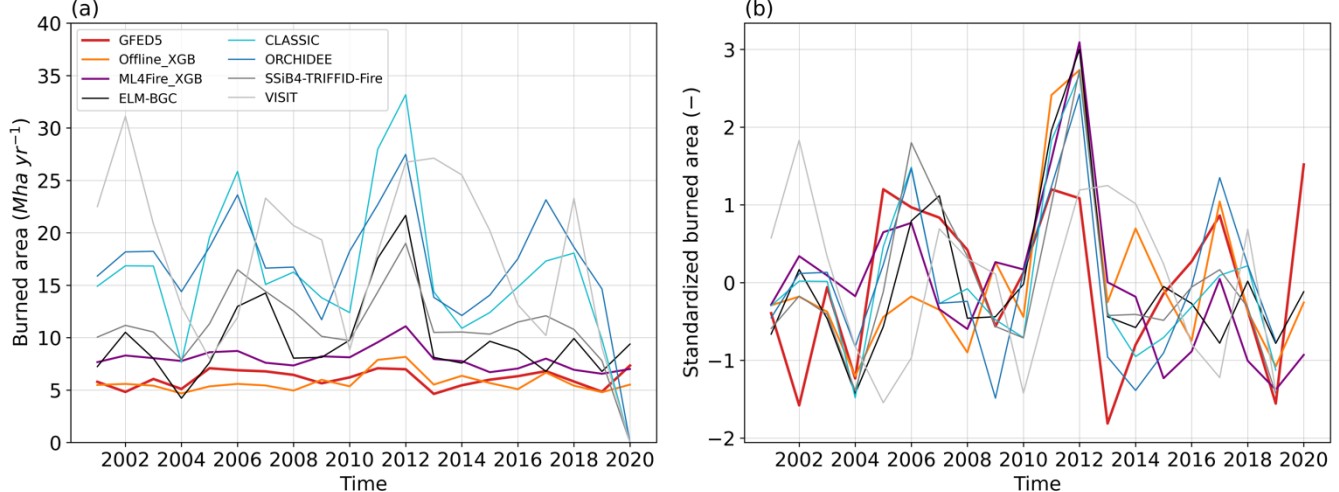

**Figure 6: Annual total burned area (Mha yr$^{-1}$). (a) Annual total and (b) standardized by removing mean and standard deviation.**

Monthly temporal variability in burned areas demonstrates significant regional differences across the eco-regions (Fig. 7). Over the entire simulation period, the ML-based models generally capture the timing of wildfires across the CONUS with a temporal correlation coefficient greater than 0.5 (p < 0.01), whereas the process-based models exhibit a correlation of only 0.3 (p > 0.01). The ML-based models also effectively capture the temporal variability across the eco-regions, although there is a slight decrease in the ML4Fire-XGB in the Great Plains and EUS. This decrease is likely related to the fire-vegetation feedback, which alters the fuel condition differently from the training set. In contrast, the process-based models show comparable correlations as the ML-based models in the WUS but fail to accurately predict burned area temporal variations in the Great Plains and EUS. Again, climatic factors play a dominant role in shaping the temporal variability of BAF in the WUS, while human activities largely influence the BAF in the Great Plains and EUS. Process-based models tend to better describe responses of fuel load and combustibility to climate than responses of fire ignition and suppression to human activities.

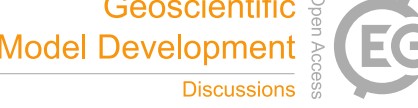



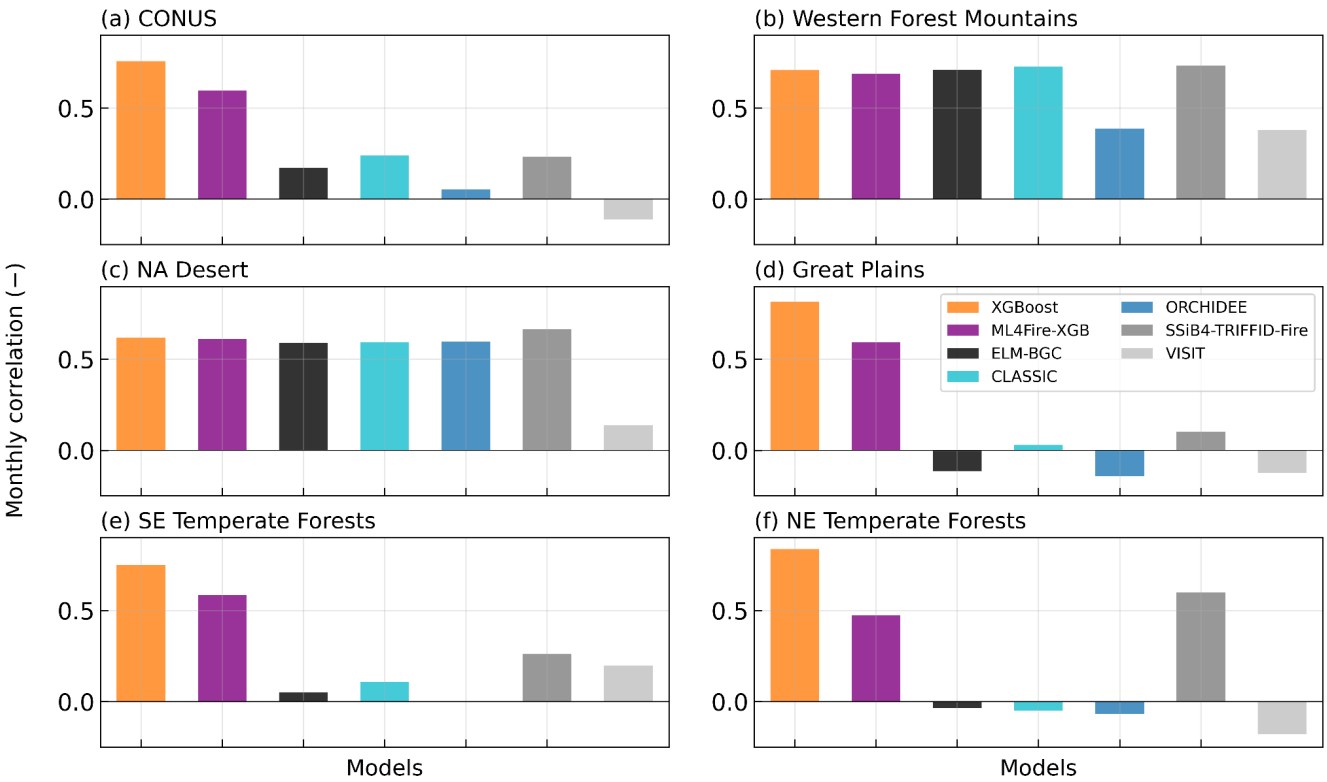

**Figure 7: Monthly correlation coefficient between simulations and GFED5 over each eco-region.**

### 3.3 Evaluation of the burned area responses to climate change

Among many other factors, the rising temperature has led to a significantly increased burned area in the WUS (Parks and Abatzoglou, 2020). The results presented in the preceding section establish the ML4Fire-XGB model as a robust tool for predicting wildfire dynamics. In this section, we examine the model responses to rising temperature by comparing the difference between simulations with/without the increasing trends in temperature. The 20-year mean difference and relative difference to the 2001-2005 mean simulated by ELM-BGC and ML4Fire-XGB are compared (Fig. 8 and Table 2). As a reference, GFED5 shows that 2001-2020 mean burned area over the COUNS increases by 0.35 Mha yr$^{-1}$ (6%) compared to the 2001-2005 mean. Both models simulate overall increased burned areas over fire-prone regions, with a comparable increase over the CONUS, 0.77 Mha yr$^{-1}$ (5%) for ELM-BGC and 0.36 Mha yr$^{-1}$ (4%) for ML4Fire-XGB. The largest increases are found over EUS, with over 25% and 7% relative increases for ELM-BGC and ML4Fire-XGB, respectively. The larger increase in EUS in ELM-BGC may originate from overpredicting the climate dependence of burned areas while overlooking the role of human activities. ELM-BGC also estimates a larger (5%) than ML4Fire-XGB (2%) over the Great Plains. Changes in the NA Desert and Great Plains are comparable in the two models. Both models estimate a 6% increase in the WUS.





**Table 2: Raising temperature induced burned area change. The relative change is regarding to 2001-2005 mean.**

|  | ELM-BGC |  | ML4Fire-XGB |  |
| --- | --- | --- | --- | --- |
|  | Mean (Mha) | Relative change (%) | Mean (Mha) | Relative change (%) |
| Western Forest Mountains | 0.13 | 5.94 | 0.05 | 5.68 |
| NA Desert | 0.02 | 2.31 | 0.04 | 4.97 |
| Great Plains | 0.11 | 4.85 | 0.09 | 2.30 |
| SE Temperate Forests | 0.43 | 25.54 | 0.15 | 7.02 |
| NE Temperate Forests | 0.07 | 26.34 | 0.03 | 7.66 |
| CONUS | 0.77 | 4.79 | 0.36 | 4.45 |

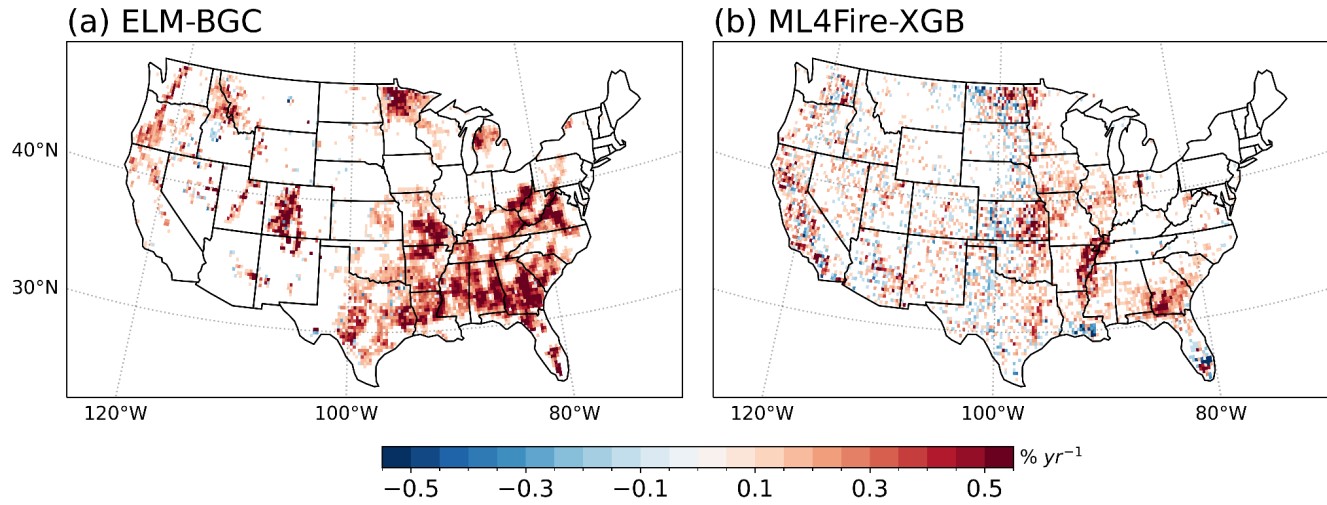

**Figure 8: 2001-2020 mean burned area fraction due to rising temperature.**

## 4 Discussion and Conclusion

In this study, we present a hybrid framework that integrates a pre-trained ML wildfire model in an Earth system model. We tailor an XGBoost wildfire model using input variables that are simulated by ELM-BGC to predict wildfires. Then the ML4ESM framework is adopted to couple the pre-trained XGBoost wildfire model with ELM, forming the ML4Fire-BGC. The ML-based wildfire models are comprehensively evaluated against observations and compared with five state-of-the-art process-based wildfire models. The offline-XGB and ML4Fire-XGB largely reduce the biases in mean burned area predictions over key regions such as WUS, where process-based models tend to overestimate burned areas by 1.5–16 times. Spatially, the





offline-XGB achieves a pattern correlation of 0.96 (p<0.01), and 0.70 (p<0.01) when integrated with ELM. The reduced
performance is caused by the fire-vegetations interaction in the coupled model. Temporally, the ML-based models accurately
capture the timing of wildfires across CONUS with a temporal correlation greater than 0.5 (p<0.01), significantly higher than
the 0.3 correlation produced by process-based models.
In the comparison of fire seasons, the ML-based models effectively capture the seasonal timing of wildfires across various
ecoregions. In the WUS, the fire season typically spans from early summer to late fall, primarily influenced by dry conditions
and high temperatures. Both ML-based and most of the process-based models simulate the July to October fire season, aligning
closely with observations. In contrast, human activities and prescribed burns significantly influence fire dynamics in the Great
Plains and EUS. Offline-XGB well reproduces the fire seasons in those regions, while ML4Fire-XGB shows degraded
performances, overperforming all process-based models. This discrepancy may arise by the potential biases in the factors
generated by ELM, which can impact the accuracy of online predictions. However, the ML-fired process exhibits high accuracy,
as demonstrated by the Offline-XGB model, making it a reliable tool for evaluating the fired area under different warming
scenarios.
The analysis of rising temperature sensitivity experiments indicates that warming is a major driver of the increased burned
area observed in recent decades. A comparison of the ELM-BGC and ML4Fire-XBG results with the total increase documented
in GFED5 suggests that ML4Fire-XBG accurately captures the burned area responses to climate variability. Consequently,
ML4Fire-XBG is well-suited for studies attributing changes in burned area to various factors.
The CLM-Li fire model (Li et al., 2012) is incorporated into both ELM and SSiB4-TRIFFID-Fire and has been partially
used in CLASSIC (Melton and Arora, 2016). Consequently, similar performance is observed among these models, while
CLASSIC showing a larger overestimation. VISIT adopts the Thonicke et al. (2001), a semi-empirical fire model and has not
been well calibrated since coupling. This may explain the poorer benchmarking results compared to other models in this study.
A fire model SPITFIRE with higher complexity (Thonicke et al., 2010), has been coupled with ORCHIDEE. Although
SPITFIRE is able to simulate both burned area and fire intensity and consider the impacts depending on fire regimes (e.g., fire
duration and flame height), plant traits (bark thickness and crown height), it does not outperform other fire models in regard
to burned area simulation (Hantson et al., 2020). With more sophisticated parametrization and fire parameters introduced,
more observational analyses are required to understand the mechanism behind and to constrain the parametric uncertainty. It
is noteworthy that parameters involved in wildfire prediction are calibrated to align with the research interests of the institute
developing and managing these models. Fine-tuning these parameters and advancing the physical understanding of wildfire
processes for the CONUS hold the potential to improve model performance (Huang et al., 2020).
The development and application of ML4Fire-XGB represent a significant step forward in our ability to model wildfire
dynamics in regions with complicated interactions between fires, ecosystems, climate, and human activities, bypassing the
explicit understanding of physical processes. By incorporating ML wildfire parameterization into a land surface model, we
address the critical need for enhanced predictive capabilities at subseasonal to seasonal scales. Meanwhile, the predictability
can adapt to the evolving nature of fire regimes under climate change. This research not only contributes to the scientific



community's understanding of fire-ecosystem-climate interactions but also provides a practical tool for policymakers and
resource managers engaged in wildfire preparedness and response.

**Author contribution**

Research conceptualization, paper preparation, and analysis were performed by YL and HH. ELM configuration and set
up was supported by DX. The hybrid model coupling framework was first developed by TZ. The GFED5 data was provided
by YC. The ML fire model development was assisted by SSW. All authors contributed to the paper edits and technical review.

**Acknowledgments**

This research has been supported by the Earth and Biological Sciences Directorate (EBSD)'s Laboratory Directed Research
and Development (LDRD) Program at Pacific Northwest National Laboratory (PNNL). D.X. was supported by the Earth
System Model Development program area of the U.S. Department of Energy, Office of Science, Office of Biological and
Environmental Research as part of the multiprogram, collaborative integrated Coastal Modeling (ICoM) project. PNNL is
operated by DOE by the Battelle Memorial Institute under contract DE-A05-76RL0 1830. The work at LLNL was performed
under the auspices of the US Department of Energy by the Lawrence Livermore National Laboratory under Contract DE-
AC52-07NA27344. Research activity at BNL was under the Brookhaven National Laboratory contract DE-SC0012704.

**Conflict of Interest**

The authors declare that they have no conflict of interest.

**Data Availability**

Data and scripts used to generate results in this study are publicly available at PNNL's DataHub
(https://doi.org/10.25584/2424127). The Fortran-Python interface (ML4ESM) for developing ML parameterizations is
archived at https://doi.org/10.5281/zenodo.11005103 (Zhang et al., 2024). The E3SM v2.1 (including ELM v2.1) is available
at    https://doi.org/10.11578/E3SM/dc.20230110.5 and https://github.com/E3SM-Project/E3SM/releases/tag/v2.1.0 (E3SM
Project,    2023).    The    modified    ELM    v2.1    (including    the    XGBoost    ML    fire    model)    is    available    at
https://doi.org/10.5281/zenodo.13358187 (Liu et al., 2024).



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
