# Peer review of "ML4Fire-XGBv1.0: Improving North American wildfire prediction by integrating a machine-learning fire model in a land surface model"

_Geoscientific Model Development, 2024_

## Referee Comment (RC1)

**Review: ML4Fire-XGBv1.0: Improving North American wildfire prediction by integrating a machine-learning fire model in a land surface model (gmd-2024-151)**

**General comments**

This manuscript builds on previous work that used climate forcing observations and vegetation model–derived vegetation outputs to build a fire model over the continental U.S. (CONUS) using the XGBoost machine learning algorithm. Here, the authors couple that fire model back into the ELM land and vegetation model, resulting in marked improvements relative to the built-in, process-based ELM fire model in terms of total burned area, its seasonal timing, and its interannual variability. There is (as expected) some decrease in performance relative to the uncoupled ML fire model, but not much. The authors also compare their ELM simulations with other process-based fire models in the FireMIP experiments. The manuscript is mostly well-structured, the figures are easy to understand, and the writing is for the most part clean and clear.

Process-based fire models are notoriously complicated and uncertain, so I am quite interested in the potential of machine learning to supplement, complement, or even replace them. However, I have serious concerns about the usefulness of the particular model system described here. I also have various less-severe but still-important concerns related to methodological and analytical issues.

To some extent these can be addressed by expanding the Discussion and adding subsections for organization. The authors should reduce the amount of space in the Discussion dedicated to reiterating already-stated results, instead only re-presenting results as needed to support new assertions. However, my fundamental concern about the usefulness of the model system presented here will require a fair amount of additional work. I thus recommend this paper be **reconsidered after major revisions**.

**Specific comments**

**Utility or "fitness for purpose" of ML-based fire model**

First, I want to outline what this concern is *not* about. There is a long-standing philosophical question of whether an empirical model can ever be trusted outside the time period and/or environmental conditions in which it was trained. Climate change and socioeconomic developments are expected to introduce never-before-seen combinations of environmental conditions and human behavior, requiring extrapolation. Some people considering this issue conclude that only process-based models are useful. However, I'm

*not* making that argument right now—my mindset is that process-based models are also imperfect, so empirical and ML models can be useful as well.

My concern is more that methodological issues in this paper make it so that I'm not sure of the usefulness of *this particular* model system. There are two main reasons for this.

First, there is not actually one "offline-XGB" fire model, but rather twenty—one for each year. The authors did this in an attempt to avoid "overfitting." Their use of that word doesn't fit with how I understand it, so I interpreted it as them avoiding training and testing their model on the same data. This is an important goal, but by choosing to do it this way, there is no single model presented that could be used for years outside 2001–2020. Instead, randomly excluding (e.g.) 20% of gridcell-years and building one model based on the rest would allow the construction of one canonical model that could be used for prognostic simulations. (In addition, it's unclear whether the authors mask low-fire gridcells only in training or also in simulations—if the latter, prognostic simulations would of course always predict zero burned area there.) This means that the ELM+offline-XGB model isn't actually useful as a predictive tool, contrary to the authors' assertions (e.g., P14 L18–19, P16–17 L72–76).

So if the presented model isn't useful for prediction, can it help us understand anything about the drivers of present-day burned area and its trends (as the authors try with the detrended-temperature experiment)? Unfortunately, the answer to that question is also no, because it's trained on unreliable model outputs of vegetation biomass, composition, and dynamics. This is a real concern with highly-tuned models, including ML models, which can end up being "right for the wrong reason," using one process to compensate for another that's poorly-represented. While observational data are also imperfect, an ML model trained only on observations—especially one designed with explainability in mind— would be more trustworthy when it comes to examining the influence of different drivers.

I think the authors could resolve this usefulness issue by building one canonical offline-XGB fire model, enabling its use in prognostic simulations. I don't really have a problem with this being trained on ELM-simulated vegetation data; yes, that will mean compensation of ELM's biases, but that can happen in pure process-based models anyway. However, it does mean that the detrended-temperature analysis should be removed from the paper, unless both the ML and process-based models' temperature responses are compared to a purely-observation-based analysis. Removing that analysis would be fine for me, as I find it somewhat extraneous.

**Methodological questions**

- Sect. 2.1.2:
    - This should be expanded to include a brief summary of how the "pretrained" model worked—enough for the reader to understand what "the large-scale patterns" are without having to consult Wang et al. (2021).

- It's also a bit confusing to say you're using the "pretrained" model, but then you change how it works and retrain it for each year from 2001–2020. Was "pretrained" supposed to mean that it's being trained offline, i.e., before being coupled?
- Excluding low-burning gridcells: Is this just in training, or are they also masked in simulations? If the latter, then the ELM-BGC and FireMIP outputs should also be masked.
- Analysis of effect of rising temperature: Why is ML4Fire-XGB included but not offline-XGB? The latter is more what this paper is actually about.
- It's unclear what FireMIP outputs you used. P6 L43: In addition to Rabin et al. (2017), the FireMIP phase 1 burned area publication should also be cited. This might be Hantson et al. (2020, doi:10.5194/gmd-13-3299-2020), but the models chosen here aren't all present in that publication. Hopefully Rabin et al. (2017) describes the simulation protocol for the models whose output you've chosen to compare; if not, a different publication that includes the protocol should be cited.

**Inconsistency of comparisons**

- This manuscript compares ML-based model performance against GFED5—were the process-based models parameterized against that? Probably not, because it's pretty new. GFED5 has 61% more burned area than GFED4s (which, the process-based fire models may have been calibrated against GFED4 or even 3). Although I'm not sure how much the increase was in CONUS. This should all be explored in the Discussion.
- I'm not sure exactly what FireMIP simulations you used, but they almost certainly used different climate, lightning, population density, and/or GDP inputs from the ELM simulations here (as well as the uncoupled ML4Fire-XGB training and usage).
- After reading the Results, the fact that the process-based models all overestimate burned area in CONUS *despite (probably) having been calibrated against the (probably) lower CONUS burned area observations* suggests that the process-based models are extra wrong!

**FireMIP models**

- Why were only those four FireMIP models chosen? This question is especially important because, as you note, two of them share (to different degrees) code derived from the same fire model used in ELM. (It was not great to learn that only in the Discussion, by the way—this is an important caveat that should have been highlighted before or perhaps in the Results section.)
- FireMIP does its own benchmarking at the global scale. If only choosing a few models, their performance in that global benchmarking should be discussed. That would help contextualize your CONUS results.
- P16 L60–61: "VISIT adopts the Thonicke et al. (2001), a semi-empirical fire model and has not been well calibrated since coupling." According to whom? Or is this just speculation?

**Agricultural burning**

- It seems like crop vegetation patches and/or burning are included in your model training and analyses? This is worth mentioning, because some process-based fire models exclude crop burning, the detection of which was a major development in GFED5. And crops had by far the largest CONUS (GFED region TENA) burned area in GFED5 (Chen et al., 2023, Table 3), although I'm not sure how much they contributed to the increase from GFED4s. Please discuss.
- Some of the FireMIP models you chose might have also excluded pasture burning; see Table S3 in Rabin et al. (2017), although that information might not apply to the versions of the models in the FireMIP simulations you chose (see my comment about P6 L43). Indeed, you acknowledge that models might not include cropland or pasture burning at P10 L50–53.
- P10 L52–53:
  - Citations should be provided for none of the models having cropland fire on.
  - "That says all vegetation models treat pastures as natural grasslands." (a) What is "That" referring to? (b) Citations? (c) For fire only, or are these also not grazed?
- P10 L53–55: "This may explain the significant overestimation of burned areas in ORCHIDEE as the SPITFIRE fire module has a much higher flammability in natural grasslands compared to woody plants." That suggestion implies that grass isn't in reality more flammable than woody plants, which I don't think is supported by evidence. Perhaps change this to something about grass being TOO much more flammable than woody plants.
- P10 L55–57:
  - Clarify that "fuel properties" includes amount as well as physical (e.g. bulk density) and chemical characteristics.
  - Management should also be mentioned here, both in terms of grazing (impacts on fuel load and plant community composition) as well as prescribed fire.
- P12 L75–84:
  - The fire model in CLM (which ELM is based on) includes crop fires. Are those not simulated in ELM? Or is their area just low (or even zero) in the Great Plains relative to other types of fire?
  - "expect" should be "except".
- P12 L85–88: This sentence should be expanded to explicitly mention and cite process-based models that do have managed crop and/or pasture burning.

**Minor comments**

- I would probably remove "North American" from the title, since this work is actually limited to the continental US—well less than half of North America.
- Title says "ML4Fire-XGBv1.0," but "ML4Fire-XGB" is used in the paper only to refer to the uncoupled ML-based fire model(s). The real development in the paper is really more about coupling the ML fire models with ELM. I'd suggest changing the model/version number in the title to something like "ELM2.1-XGBfire1.0".

- Line numbers seem to just show the last two digits; please fix in revision.
- Various places: "COUNS" typo.
- P2 L27–29: "[C]limate change has contributed to a 16% increase in the global burned area over the past two decades, while human influences, including ignition and suppression, have reduced by 27%." Second part is sort of ambiguous. Has the strength of the human influences decreased by 27%, or have human influences caused burned area to be reduced by 27%?
- P3 L67–69: "The corresponding changes in fire dynamics may shift the vegetation species distribution from those originally low in resistance to wildfire to those in high resistance or even benefiting from regular fire occurrence (Rogers et al., 2015; Huang et al., 2024)." Since you're using big leaf, you're not getting that—this should be discussed.
- P4 L92–93:
    - Is suppression not also a function of population density (in addition to GDP)?
    - Per-capita GDP, no?
- P4 L04: Worth pointing out that "competition" in ELM (without FATES turned on, that is) is limited to competition for soil resources, not light.
- P5 L16: "To reduce overfitting, we build a separate ML model for each year from 2001 to 2020 using the remaining 19 years' data." Confused me for a while. Suggested revision in bold: "To reduce overfitting, we build a separate ML model for each year from 2001 to 2020 using the **data from the other 19 years in that period**."
- P6 L40–41: "significant" should be "important" or something similar; "while all zero burned areas" seems to be an incomplete thought.
- P6 L42–48: Also mention here that you'll be looking at ELM-BGC outputs.
- P6 L62: Missing degree symbol at "0.25x0.25".
- P6 L62: How were the datasets resampled? Nearest-neighbor?
- Table 1: GDP and population density citations don't match text at P6 L61–62.
- P8 L97: "XBG" should be "XGB".
- Fig. 3:
    - What is gray?
    - Would it be more useful to have the ecoregions overlaid on this map instead of state boundaries? I could see an argument either way.
    - Please add text boxes with each model's $R_p$ (including asterisks to show significance level) and bias scores. As it is now, some models don't have their scores listed anywhere in the text/figures/tables.
- P9 L22–23: "the performance holds," but it actually worsens, as the next sentence says.
- P10 L52: "intermodal" should be "intermodal".
- P10–11 L57–59: If fires in this region are managed by prescribed burning, we actually *shouldn't* expect the process-based models to do well there, since they don't account for prescribed burning. This result is thus somewhat surprising.
- P11 Fig. 4: To reflect interannual variability, add uncertainty bars and/or change the red line to a shaded region.
- P13 L93: "accounting for" should be "representing"; "largely" should be "greatly"; "in" should be "from".

- P13 L99: "IVA" should be "IAV".
- P13 L11–13: "Again, climatic factors play a dominant role in shaping the temporal variability of BAF in the WUS, while human activities largely influence the BAF in the Great Plains and EUS. Process-based models tend to better describe responses of fuel load and combustibility to climate than responses of fire ignition and suppression to human activities." Citations needed for these statements.
- P15, Fig. 8: What is white?
- Two different reference lists? The first one ends and the second begins on P19. They're not the same, either, with at least one reference (Donovan et al., 2020) missing from the first.

---

## Author Comment (AC1)

We sincerely appreciate the referees for their valuable and insightful comments on our manuscript. The feedback is instrumental in enhancing the quality and clarity of our research. These comments are not only valuable but also serve as a critical resource for improving various aspects of our article, including methodology, data interpretation, and overall presentation. We have taken each comment seriously and conducted a thorough review of our manuscript to ensure that we comprehensively address all concerns raised by the referees.

This response document provides a detailed account of the changes implemented in relation to each specific comment from the referee. For ease of reference, referee comments are presented in black, while author responses are highlighted in blue, with modifications to the manuscript in italic font. The line numbers correspond to the clean version of the revision.

**General comments**

This manuscript builds on previous work that used climate forcing observations and vegetation model–derived vegetation outputs to build a fire model over the continental U.S. (CONUS) using the XGBoost machine learning algorithm. Here, the authors couple that fire model back into the ELM land and vegetation model, resulting in marked improvements relative to the built-in, process-based ELM fire model in terms of total burned area, its seasonal timing, and its interannual variability. There is (as expected) some decrease in performance relative to the uncoupled ML fire model, but not much. The authors also compare their ELM simulations with other process-based fire models in the FireMIP experiments. The manuscript is mostly well-structured, the figures are easy to understand, and the writing is for the most part clean and clear.

Process-based fire models are notoriously complicated and uncertain, so I am quite interested in the potential of machine learning to supplement, complement, or even replace them. However, I have serious concerns about the usefulness of the particular model system described here. I also have various less-severe but still-important concerns related to methodological and analytical issues.

To some extent these can be addressed by expanding the Discussion and adding subsections for organization. The authors should reduce the amount of space in the Discussion dedicated to reiterating already-stated results, instead only re-presenting results as needed to support new assertions. However, my fundamental concern about the usefulness of the model system presented here will require a fair amount of additional work. I thus recommend this paper be reconsidered after major revisions.

Thank you for your thorough review and valuable insights. We agree that the discussion section would benefit from additional subsections to better organize key points and reduce repeated results. We will reorganize this section, highlighting methodological implications and comparison insights while minimizing redundancy. Additionally, we fully agree with your perspective on the roles of process-based models and machine learning approaches. Process-based models provide critical insights into mechanistic processes, while machine learning approaches offer opportunities to capture complex patterns that may be challenging to model directly. We believe

combining these methods can enhance predictive capabilities and address uncertainties effectively. We have reorganized the Discussion section to avoid repeating contents and extend the discussions.

**Specific comments**

*Utility or "fitness for purpose" of ML-based fire model*

First, I want to outline what this concern is not about. There is a long-standing philosophical question of whether an empirical model can ever be trusted outside the time period and/or environmental conditions in which it was trained. Climate change and socioeconomic developments are expected to introduce never-before-seen combinations of environmental conditions and human behavior, requiring extrapolation. Some people considering this issue conclude that only process-based models are useful. However, I'm *not* making that argument right now—my mindset is that process-based models are also imperfect, so empirical and ML models can be useful as well.

My concern is more that methodological issues in this paper make it so that I'm not sure of the usefulness of *this particular* model system. There are two main reasons for this.

First, there is not actually one "offline-XGB" fire model, but rather twenty—one for each year. The authors did this in an attempt to avoid "overfitting." Their use of that word doesn't fit with how I understand it, so I interpreted it as them avoiding training and testing their model on the same data. This is an important goal, but by choosing to do it this way, there is no single model presented that could be used for years outside 2001–2020. Instead, randomly excluding (e.g.) 20% of gridcell-years and building one model based on the rest would allow the construction of one canonical model that could be used for prognostic simulations. (In addition, it's unclear whether the authors mask low-fire gridcells only in training or also in simulations—if the latter, prognostic simulations would of course always predict zero burned area there.) This means that the ELM+offline-XGB model isn't actually useful as a predictive tool, contrary to the authors' assertions (e.g., P14 L18–19, P16–17 L72–76).

We fully understand your concern. The concept of splitting data by years and building a separate model for each year was to ensure all the predictions were not using data that the machine learning model had seen during its training process. However, we acknowledge that this approach oversights the usefulness of this model. Regarding masking low-fire grid cells, we only applied the mask during the training process.

In the revision, we have trained a canonical model using the random split approach, 80% for training and 20% for validation. The canonical model retains the performance in both offline and coupled models, and this single model will be useful in prognostic simulations. For more details, please see the responses to the specific comment.

So if the presented model isn't useful for prediction, can it help us understand anything about the drivers of present-day burned area and its trends (as the authors try with the detrended-temperature experiment)? Unfortunately, the answer to that question is also no,

because it's trained on unreliable model outputs of vegetation biomass, composition, and dynamics. This is a real concern with highly-tuned models, including ML models, which can end up being "right for the wrong reason," using one process to compensate for another that's poorly-represented. While observational data are also imperfect, an ML model trained only on observations—especially one designed with explainability in mind— would be more trustworthy when it comes to examining the influence of different drivers. I think the authors could resolve this usefulness issue by building one canonical offline-XGB fire model, enabling its use in prognostic simulations. I don't really have a problem with this being trained on ELM-simulated vegetation data; yes, that will mean compensation of ELM's biases, but that can happen in pure process-based models anyway. However, it does mean that the detrended-temperature analysis should be removed from the paper, unless both the ML and process-based models' temperature responses are compared to a purely-observation-based analysis. Removing that analysis would be fine for me, as I find it somewhat extraneous.

We appreciate your understanding of training the ML model with ELM-simulated vegetation data and your insights about model error compensation. We acknowledge the limitation of attribution analysis using a highly-turned model. In the revision, the analysis and discussion on the detrended temperature have been removed.

*Methodological questions*

• Sect. 2.1.2:

This should be expanded to include a brief summary of how the "pretrained" model worked—enough for the reader to understand what "the large-scale patterns" are without having to consult Wang et al. (2021).

We have added the following texts to expand the introduction on the pretrained XGBoost model (Lines 115-123).

*In this study, we adapted the XGBoost algorithm used by Wang et al (2021) to develop an offline ML fire model using variables directly provided by ELM at each grid cell. Wang et al. (2021) integrated large-scale meteorological patterns alongside local weather, land surface properties, and socioeconomic data to enhance the prediction of burned areas. The large-scale patterns were identified using singular value decomposition (SVD) to capture influential atmospheric conditions that develop over days to weeks and cumulatively impact the monthly burned area. The feature importance analysis in their study noted that while large-scale patterns improved prediction, however, they played a secondary role. Therefore, we exclude the large-scale patterns from predictors without significantly affecting the model accuracy. Hereafter the uncoupled XGBoost fire model is referred to as offline-XGB.*

It's also a bit confusing to say you're using the "pretrained" model, but then you change how it works and retrain it for each year from 2001–2020. Was "pretrained" supposed to mean that it's being trained offline, i.e., before being coupled?

Thanks for raising this great point. In the revision, we modified this sentence to:

*In this study, we adapted the XGBoost algorithm used by Wang et al (2021) to develop an offline ML fire model using variables directly provided by ELM at each grid cell.*

And

*Hereafter the uncoupled XGBoost fire model is referred to as offline-XGB.*

Excluding low-burning gridcells: Is this just in training, or are they also masked in simulations? If the latter, then the ELM-BGC and FireMIP outputs should also be masked.

The low-burning grid cell mask is only applied in the training process. We have clarified this in the revision.

• Analysis of effect of rising temperature: Why is ML4Fire-XGB included but not offline-XGB? The latter is more what this paper is actually about.

In the original manuscript, we used ML4Fire-XGB in the rising temperature experiment to account for the temperature effect on vegetation growth and its consequence on the burned area. However, we agree with the reviewer's inspection of this analysis, and we have removed the relevant analysis and discussion.

• It's unclear what FireMIP outputs you used. P6 L43: In addition to Rabin et al. (2017), the FireMIP phase 1 burned area publication should also be cited. This might be Hantson et al. (2020, doi:10.5194/gmd-13-3299-2020), but the models chosen here aren't all present in that publication. Hopefully Rabin et al. (2017) describes the simulation protocol for the models whose output you've chosen to compare; if not, a different publication that includes the protocol should be cited.

Thank you for pointing out the confusion. In our manuscript, we used outputs from the latest FireMIP models, i.e., the FireMIP Phase II or the ISIMIP-Fire sector (ISIMIP3a) (https://protocol.isimip.org/#/ISIMIP3a/fire). The ISIMIP3a experimental design follows the protocol outlined in Rabin et al. (2017), with all models using a common set of climate and socioeconomic (land-use, GDP etc) data provided by ISIMIP3a. We opted for ISIMIP3a model outputs due to the updates in fire models implemented and a longer simulation period post-2000 in this phase. Although there is no specific protocol paper for ISIMIP3a, two publications have recently been made available (Burton et al., 2024; Park et al., 2024). At the time of our study, only four models had uploaded results, but we have now updated our analysis to include outputs from all seven ISIMIP3a models, as detailed in Burton et al. (2024).

Reference:

Burton, C., Lampe, S., Kelley, D. I., Thiery, W., Hantson, S., Christidis, N., Gudmundsson, L., Forrest, M., Burke, E., Chang, J., Huang, H., Ito, A., Kou-Giesbrecht, S., Lasslop, G., Li, W., Nieradzik, L., Li, F., Chen, Y., Randerson, J., Reyer, C. P. O., and Mengel, M.: Global burned area increasingly explained by climate change, Nat Clim Change, 10.1038/s41558-024-02140-w, 2024.

Park, C. Y., Takahashi, K., Fujimori, S., Jansakoo, T., Burton, C., Huang, H., Kou-Giesbrecht, S., Reyer, C. P. O., Mengel, M., Burke, E., Li, F., Hantson, S., Takakura, J., Lee, D. K., and Hasegawa, T.: Attributing human mortality from fire PM2.5 to climate change, Nat Clim Change, 10.1038/s41558-024-02149-1, 2024.

*Inconsistency of comparisons*

• This manuscript compares ML-based model performance against GFED5—were the process-based models parameterized against that? Probably not, because it's pretty new. GFED5 has 61% more burned area than GFED4s (which, the process-based fire models may have been calibrated against GFED4 or even 3). Although I'm not sure how much the increase was in CONUS. This should all be explored in the Discussion.

We appreciate the reviewer's concern, particularly as we developed one of the process-based models participating in ISIMIP3a, and GFED4s served as the reference dataset for global fire calibration in these models. For the ML model training, we chose GFED5 because it has been shown to better capture small fires compared to earlier datasets (Chen et al., 2023; Roteta et al., 2019), which often under-represent prevalent agricultural fires in the Central U.S. Additionally, GFED5 is now used as the reference dataset in the latest FireMIP/ISIMIP3a publication (e.g., Burton et al., 2024). While differences in magnitude exist between GFED5 and GFED4s in burned area estimates within CONUS, these datasets also share common features. The spatial correlations of GFED4s and FireCCI5.1 against GFED5 are over 0.66. In the revision, we have added a new figure (Figure 3, also see below) comparing GFED5, GFED4s, and FireCCI5.1. The process-based models indeed face challenges in accurately predicting burned areas over CONUS, even when evaluated against GFED4s or FireCCI5.1.

[Figure]

*Figure 3: Observed burned area fraction (% yr-1). (a) GFED5 (2001-2019), (b) GFED4s (2001-2016), and (c) FireCCI5.1 (2001-2019). The numbers indicate the mean (M) burned area fraction and burned area (in Mha) in brackets for each dataset. The pattern correlation (R) against GFED5 is also shown, with an asterisk (\*) denoting significance at the 0.01 level. Black contours outline the ecoregions.*

Reference

Chen, Y. et al. Multi-decadal trends and variability in burned area from the fifth version of the Global Fire Emissions Database (GFED5). Earth Syst Sci Data 15, 5227–5259 (2023).

Roteta, E., Bastarrika, A., Padilla, M., Storm, T. & Chuvieco, E. Development of a Sentinel-2 burned area algorithm: Generation of a small fire database for sub-Saharan Africa. Remote Sens Environ 222, 1–17 (2019).

• I'm not sure exactly what FireMIP simulations you used, but they almost certainly used different climate, lightning, population density, and/or GDP inputs from the ELM simulations here (as well as the uncoupled ML4Fire-XGB training and usage).

This study uses model simulations from the ISIMIP3a (FireMIP phase II). In our ELM-BGC and ELM2.1-XGBfire1.0 (the coupled ELM and XGB fire model) simulations, we adopted the same lightning, $CO_2$, population density, and GDP data used in ISIMIP3a, with the exception of the climate forcing data. To focus on fires in CONUS, we applied the hourly NLDAS climate forcing at a spatial resolution of 0.25º, rather than the daily GSWP3-W5E5 forcing at 0.5º used in ISIMIP3a. This different reanalysis data source and differences in the spatial and temporal resolutions of the climate forcing could contribute to variations in burned area predictions.

Besides ISIMIP3a models, we also conducted ELM-BGC (with built-in process-based fire model) simulations driven by the same set of climate, lightning, and socioeconomic forcing data as used to drive the coupled model ELM2.1-XGBfire1.0. The results show that the burned area simulation in ELM-BGC remains unsatisfactory, indicating that changes in climate forcing alone do not account for all limitations in burned area simulations in process-based models (at least for ELM-BGC). We have added the following discussion to the revised manuscript to clarify this point (Lines 360-363).

*The ISIMIP3a models were driven by daily GSWP3-W5E5 forcings at a 0.5º spatial resolution. Differences in forcing data could lead to variations in burned area predictions. However, since both ELM-BGC and ELM2.1-XGBfire1.0 are driven by the same set of forcings, this suggests that limitations in physical processes may significantly hinder the performance of process-based models.*

• After reading the Results, the fact that the process-based models all overestimate burned area in CONUS despite (probably) having been calibrated against the (probably) lower CONUS burned area observations suggests that the process-based models are extra wrong!

The reviewer's observation is correct. The estimation over the CONUS in GFED5 is 114% larger than GFED4s (Figure 3 in the revision). As we discussed in the manuscript, the process-based models often focus on the globe or fire-prone regions such as African savannas, where fire regimes can be distinct from the CONUS. We believe process-based model performance over the CONUS can be improved with parameter calibration and an advanced understanding of the missing physics.

*FireMIP models*

• Why were only those four FireMIP models chosen? This question is especially important because, as you note, two of them share (to different degrees) code derived from the same fire model used in ELM. (It was not great to learn that only in the Discussion, by the way—this is an important caveat that should have been highlighted before or perhaps in the Results section.)

The four models were obtained from FireMIP phase II (ISIMIP3a) (https://protocol.isimip.org/#/ISIMIP3a/fire). By the time this research was performed, only these four models were available. However, we have now updated our analysis to include a total of seven ISIMIP3a models used in the latest ISIMIP3a benchmarking study (Burton et al. 2024). For instance, the spatial map comparison (Figure 4 in revision) has been updated as follows. The different fire models used by each ISIMIP3a model are now described in the method section 2.2.1.

[Figure]

*Figure 4. Same as Figure 3, but shows model outputs. The bias (B) against GFED5 is indicated.*

Reference:

Burton, C., Lampe, S., Kelley, D. I., Thiery, W., Hantson, S., Christidis, N., Gudmundsson, L., Forrest, M., Burke, E., Chang, J., Huang, H., Ito, A., Kou-Giesbrecht, S., Lasslop, G., Li, W., Nieradzik, L., Li, F., Chen, Y., Randerson, J., Reyer, C. P. O., and Mengel, M.: Global burned area increasingly explained by climate change, Nat Clim Change, 10.1038/s41558-024-02140-w, 2024.

• FireMIP does its own benchmarking at the global scale. If only choosing a few models, their performance in that global benchmarking should be discussed. That would help contextualize your CONUS results.

Thank you for the suggestion. The global benchmarking performance of the models was thoroughly discussed by Burton et al. (2024). To avoid redundancy, we focused our analysis on the CONUS rather than repeating the global evaluation. In the revision, we have included all seven models and discussed their performance over the CONUS.

• P16 L60–61: "VISIT adopts the Thonicke et al. (2001), a semi-empirical fire model and has not been well calibrated since coupling." According to whom? Or is this just speculation?

This sentence has been written as follow:

*VISIT employs the semi-empirical fire model developed by Thonicke et al. (2001), primarily to predict fire emissions. Burned area is calculated annually without accounting for specific ignition mechanisms. This annual burned area is then distributed across each month by weighting it according to the ratio of monthly fire season length to annual fire season length.*

*Agricultural burning*

• It seems like crop vegetation patches and/or burning are included in your model training and analyses? This is worth mentioning, because some process-based fire models exclude crop burning, the detection of which was a major development in GFED5. And crops had by far the largest CONUS (GFED region TENA) burned area in GFED5 (Chen etal., 2023, Table 3), although I'm not sure how much they contributed to the increase from GFED4s. Please discuss.

The crop PFT fraction is included as a predictor in our model, and crop burning is incorporated within the GFED5 burned area data used for training. Consequently, our XGB model is capable of predicting agricultural burning patterns. This inclusion is important because crop burning constitutes 49% of the total burned area in the CONUS, as highlighted by GFED5 (Chen et al., 2023). By accounting for crop burning, our model aligns more closely with recent advancements in fire detection and provides a more comprehensive representation of fire activity across different land cover types, including agricultural areas. We have added the following discussion in Line 363-366.

*In contrast, for instance, the ML model includes the crop PFT fraction and accounts for agricultural burning in its training data, enabling it to capture agricultural burning patterns that are typically missing or underrepresented in process-based models. This inclusion is particularly significant in the CONUS, where agricultural burning constitutes 49% of the total burned area (Chen et al., 2023).*

• Some of the FireMIP models you chose might have also excluded pasture burning; see Table S3 in Rabin et al. (2017), although that information might not apply to the versions of the models in the FireMIP simulations you chose (see my comment about P6 L43). Indeed, you acknowledge that models might not include cropland or pasture burning at P10 L50–53.

Thank you for raising this point. None of these models explicitly accounts for crop (residual) fires. Most models, except JULES, consider croplands as non-burnable. JULES treats cropland similarly to natural grassland, while all other models exclude cropland from burning entirely. Fires are permitted in pastures across all models. In LPJ-GUESS-SIMFIRE-BLAZE, pastures are harvested, which results in reduced biomass and, consequently, a smaller burned area. In contrast, other models treat pastures as natural grasslands in terms of growth and fire behavior.

For more details, please refer to Extended Data Table 1 (Fire model overview) and Section 3 of the Supplementary Material in Burton et al. (2024), and Teckentrup et al., 2019.

An introduction of the current treatment of crop fire in the current ISIMIP3a models has been added in Lines 159-164.

*The representation of fires over croplands and pastures varies across models (Burton et al. 2024, Teckentrup et al. 2019). Most models, except for JULES, classify croplands as non-burnable. JULES treats croplands similarly to natural grasslands regarding fire behavior, while all other models exclude croplands from burning. Fires are allowed in pastures in all models in terms of both growth and fire behavior. In LPJ-GUESS-SIMFIRE-BLAZE, pastures are harvested, leading to reduced biomass and consequently a smaller burned area. In contrast, other models treat pastures as natural grasslands.*

Reference:

Teckentrup, L., Harrison, S. P., Hantson, S., Heil, A., Melton, J. R., Forrest, M., Li, F., Yue, C., Arneth, A., Hickler, T., Sitch, S., and Lasslop, G.: Response of simulated burned area to historical changes in environmental and anthropogenic factors: a comparison of seven fire models, Biogeosciences, 16, 3883-3910, 2019.

• P10 L52–53:

Citations should be provided for none of the models having cropland fire on.

References (Burton et al. 2024) and (Teckentrup et al. 2019) have been added. Thank you.

"That says all vegetation models treat pastures as natural grasslands." (a) What is "That" referring to? (b) Citations? (c) For fire only, or are these also not grazed?

We will clarify this in the next manuscript in Lines 276-280.

*As noted by Teckentrup et al. (2019) and Burton et al. (2024), none of the process-based models has activated the explicit cropland fire model. Fires are allowed in pastures. While LPJ-GUESS-SIMFIRE-BLAZE incorporates harvesting in pastures, reducing biomass and influencing fire dynamics, all other process-based vegetation models treat pastures as natural grasslands for both vegetation growth and fire processes.*

• P10 L53–55: "This may explain the significant overestimation of burned areas in ORCHIDEE as the SPITFIRE fire module has a much higher flammability in natural grasslands compared to woody plants." That suggestion implies that grass isn't in reality more flammable than woody plants, which I don't think is supported by evidence. Perhaps change this to something about grass being TOO much more flammable than woody plants.

Thank you for pointing this out. We agree that in ORCHIDEE, the flammability of grass might be set too high relative to that of trees, as discussed in Teckentrup et al. (2019). We have revised this sentence in the updated manuscript to clarify this point.

• P10 L55–57:

Clarify that "fuel properties" includes amount as well as physical (e.g. bulk density) and chemical characteristics.

Modified as suggested.

Management should also be mentioned here, both in terms of grazing (impacts on fuel load and plant community composition) as well as prescribed fire.

The following sentence has been added to Lines 284-285.

*Fuel management practices, such as prescribed burning and grazing, can significantly alter fire dynamics but are generally absent in current models.*

• P12 L75–84: The fire model in CLM (which ELM is based on) includes crop fires. Are those not simulated in ELM? Or is their area just low (or even zero) in the Great Plains relative to other types of fire?

Crop fires are not enabled in the version of the ELM model we used. The crop model has not been explicitly calibrated to represent crop fires in CONUS, and enabling it could introduce additional biases due to parameter uncertainties.

"expect" should be "except".

Corrected.

• P12 L85–88: This sentence should be expanded to explicitly mention and cite process-based models that do have managed crop and/or pasture burning.

To the best of our knowledge, ELM is one of the few process-based models capable of explicitly simulating crop fires; however, this feature was not enabled in our study. None of the models used here include explicit representations of pasture burning. We have added this statement to the revision.

Minor comments

• I would probably remove "North American" from the title, since this work is actually limited to the continental US—well less than half of North America.

"North American" has been removed from the title.

• Title says "ML4Fire-XGBv1.0," but "ML4Fire-XGB" is used in the paper only to refer to the uncoupled ML-based fire model(s). The real development in the paper is really more about coupling the ML fire models with ELM. I'd suggest changing the model/version number in the title to something like "ELM2.1-XGBfire1.0".

It's a great point. We have changed the title to:

*ELM2.1-XGBfire1.0: Improving wildfire prediction by integrating a machine-learning fire model in a land surface model*

• Line numbers seem to just show the last two digits; please fix in revision.

Looks like it happened when converting from Word to PDF. It has been corrected in the revision. Sorry for the inconvenience.

• Various places: "COUNS" typo.

Apologies for the typos. We have thoroughly gone through the manuscript to avoid these typos.

• P2 L27–29: "[C]limate change has contributed to a 16% increase in the global burned area over the past two decades, while human influences, including ignition and suppression, have reduced by 27%." Second part is sort of ambiguous. Has the strength of the human influences decreased by 27%, or have human influences caused burned area to be reduced by 27%?

R: We apologize for the confusion. We have rewritten this sentence as follow (Lines 27-31).

*Globally, modeling studies show that climate change since the early 1900s has contributed to a 16% increase in the total burned area; however, human activities have led to a 19% decrease, resulting in a slight net decline in burned area over the 20th century (Burton et al. 2024). In the past two decades, satellite-derived data suggest that the global total burned area has declined by over 20%, with this trend primarily attributed to human influences (Jones et al. 2022; Andela et al. 2017).*

• P3 L67–69: "The corresponding changes in fire dynamics may shift the vegetation species distribution from those originally low in resistance to wildfire to those in high resistance or even benefiting from regular fire occurrence (Rogers et al., 2015; Huang et al., 2024)." Since you're using big leaf, you're not getting that—this should be discussed.

Currently, ELM is configured in the "biogeochemistry" (BGC) model, with PFT distributions prescribed based on satellite products. We have clarified this in the discussion of the revised manuscript in lines 394-396.

• P4 L92–93: Is suppression not also a function of population density (in addition to GDP)? Per-capita GDP, no?

Thank you for pointing this out. Yes. Suppression is parameterized as a function of GDP per capita and population density in ELM. Correction has been made in the revision.

• P4 L04: Worth pointing out that "competition" in ELM (without FATES turned on, that is) is limited to competition for soil resources, not light.

We have clarified it in the revision, Lines 105-106.

*The post-fire vegetation recovery in ELM-BGC depends on the plant photosynthesis processes and PFT competition strategy for soil resources.*

• P5 L16: "To reduce overfitting, we build a separate ML model for each year from 2001 to 2020 using the remaining 19 years' data." Confused me for a while. Suggested revision in bold: "To reduce overfitting, we build a separate ML model for each year from 2001 to 2020 using the **data from the other 19 years in that period**."

This sentence has been removed since the random splitting is used to build a canonical model.

• P6 L40–41: "significant" should be "important" or something similar; "while all zero burned areas" seems to be an incomplete thought.

This sentence has been rewritten as follows (Lines 200-201). Thank you.

*This step is important to avoid feeding the ML model distinct predictor combinations that all correspond to zero burned areas, which could skew the model's learning process.*

• P6 L42–48: Also mention here that you'll be looking at ELM-BGC outputs.

The ELM-BGC has been added.

• P6 L62: Missing degree symbol at "0.25x0.25".

Corrected.

• P6 L62: How were the datasets resampled? Nearest-neighbor?

All variables are interpolated using the bilinear interpolation method for spatial and nearest-neighbor for temporal.

• Table 1: GDP and population density citations don't match text at P6 L61–62.

Corrected.

• P8 L97: "XBG" should be "XGB".

Corrected. Thank you.

• Fig. 3:

What is gray?

The gray shading has been removed in the revised manuscript.

Would it be more useful to have the ecoregions overlaid on this map instead of state boundaries? I could see an argument either way. Please add text boxes with each model's Rp (including asterisks to show significance level) and bias scores. As it is now, some models don't have their scores listed anywhere in the text/figures/tables.

We have updated Figures 3 and 4 in the revision as suggested. Please see the figures in response to your major comments.

• P9 L22–23: "the performance holds," but it actually worsens, as the next sentence says.

Thanks for pointing this out. This sentence has been changed to:

*While integrated with ELM, the performance was slightly degraded.*

• P10 L52: "intermodal" should be "intermodal".

We changed "intermodal" to "inter-model".

• P10–11 L57–59: If fires in this region are managed by prescribed burning, we actually shouldn't expect the process-based models to do well there, since they don't account for prescribed burning. This result is thus somewhat surprising.

It is a great point. Based on our conversation with local agencies, they tend to cast prescribed fires as much as possible, which effectively reduces large fires and makes the actual burned areas largely influenced by climate and fuel conditions. Since the ignition is less constrained, the burned area is mainly influenced by the fire spread which is highly related to natural forcing such as fuel and wind conditions. On the other hand, the southeastern U.S. is a lightning-prone region, which is a major source of fire ignition in models. Therefore, without prescribed burning, models simulated a high level of fire ignition due to lightning, and well capture the fire spread.

The following discussion has been added in Lines 288-290:

*Although prescribed burning as an additional ignition source is not included in the process-based models, ignition is not a limiting factor in this region due to the abundance of lightning, which provides sufficient natural ignition sources. Consequently, burned area is primarily controlled by fire spread, which is influenced by natural conditions such as fuel availability and wind, allowing the models to perform well in simulating fire dynamics.*

• P11 Fig. 4: To reflect interannual variability, add uncertainty bars and/or change the red line to a shaded region.

One-standard deviation range of the red line (GFED5) has been added to reflect the uncertainty.

[Figure]

• P13 L93: "accounting for" should be "representing"; "largely" should be "greatly"; "in" should be "from".

Corrected. Thank you.

• P13 L99: "IVA" should be "IAV".

Corrected.

• P13 L11–13: "Again, climatic factors play a dominant role in shaping the temporal variability of BAF in the WUS, while human activities largely influence the BAF in the Great Plains and EUS. Process-based models tend to better describe responses of fuel load and combustibility to climate than responses of fire ignition and suppression to human activities." Citations needed for these statements.

Citations including Kupfer et al. 2020, Chen et al. 2023, and Hantson et al. 2016 have been added.

Kupfer, J. A., Terando, A. J., Gao, P., Teske, C., and Kevin Hiers, J.: Climate change projected to reduce prescribed burning opportunities in the south-eastern United States, Int. J. Wildland Fire, 29, 764–778, 2020.

Chen, Y., Hall, J., van Wees, D., Andela, N., Hantson, S., Giglio, L., van der Werf, G. R., Morton, D. C., and Randerson, J. T.: Global fire emissions database (GFED5) burned area, https://doi.org/10.5281/ZENODO.7668423, 2023.

Hantson, S., Arneth, A., Harrison, S. P., Kelley, D. I., Prentice, I. C., Rabin, S. S., Archibald, S., Mouillot, F., Arnold, S. R., Artaxo, P., Bachelet, D., Ciais, P., Forrest, M., Friedlingstein, P., Hickler, T., Kaplan, J. O., Kloster, S., Knorr, W., Lasslop, G., Li, F., Mangeon, S., Melton, J. R., Meyn, A., Sitch, S., Spessa, A., van der Werf, G. R., Voulgarakis, A., and Yue, C.: The status and challenge of global fire modelling, Biogeosciences, 13, 3359–3375, 2016.

• P15, Fig. 8: What is white?

This figure has been removed.

• Two different reference lists? The first one ends and the second begins on P19. They're not the same, either, with at least one reference (Donovan et al., 2020) missing from the first.

Thanks for checking that! We have corrected the reference list in the revision.

---

## Author Comment (AC2)

We sincerely thank Dr. Kasoar for the valuable and insightful feedback on our manuscript. The comments provided have been instrumental in enhancing the quality and clarity of our work, serving as a critical resource for refining our methodology, data interpretation, and overall presentation. We have carefully considered each comment and have conducted a thorough revision of our manuscript to comprehensively address all concerns raised.

This response document details the specific changes made in response to each reviewer comment. For ease of reference, reviewer comments are presented in black, our responses are highlighted in blue, and modifications to the manuscript are indicated in italic font. The line numbers correspond to the clean version of the revised manuscript.

The authors present a new two-way coupling of an XGBoost machine learning fire model over the contiguous US, with the ELM land surface model, a derivative of the widely-used CLM land-surface and dynamic vegetation model, which can be run as an alternative to the process-based Li et al. scheme currently used within ELM (and CLM). The XGB fire model performs very well at reproducing the observation-based training dataset (GFED5 burned area) over the CONUS. The authors also compare against BA simulations from several process-based models, and note that agreement is better over regions of the CONUS where burned area is mainly driven by climate, and poorer over regions where there is a strong human influence on the burned area signature, e.g. due to crop and pasture fires, thereby highlighting another potential use of ML models to help identify key process that should be better represented in their process-based counterparts.

ML methods show a lot of promise when it comes to more accurately parameterising sub-grid phenomena, including wildfire prediction which is notoriously uncertain among current process-based and simple empirical models, and so I really welcome initiatives like this to interactively couple data-driven fire models with a dynamic vegetation model to provide an alternative to the existing process-based scheme, depending on the desired application. I have some high-level concerns detailed below about the current presentation of the model description and validation; hopefully most of these can be resolved through additional discussion and clarifications - and very happy to be corrected if I'm mistaken or have misunderstood anything! I also have some recommendations for additional analysis and validation that I think could be beneficial. I then finally list a few very minor technical comments/clarifications.

In terms of the big picture motivation of the paper, the main new development is the interactive coupling of the XGB fire model with the ELM land surface and dynamic vegetation model. Therefore, it seems strange that no results or analysis are presented showing the feedbacks that are possible because of this coupling. All the model outputs presented are focused purely on burned area validation - where in fact, the coupled ELM-ML4Fire-XGB model performs slightly worse than just using offline XGB, presumably due to the coupling feedbacks which influence the vegetation distribution. So as it stands, the paper doesn't really motivate why coupling these models is desirable; if you just want

burned area accuracy, it's better to use the XGB model offline.  The key benefit is presumably the feedbacks on vegetation distribution, carbon fluxes, etc.  One would imagine that the interactive vegetation distribution in ELM is improved when it's impacted by a more realistic fire distribution, or that the feedbacks on vegetation due to changing fire regimes over time are better simulated.  So, it would be nice to see some results showing how vegetation-related variables are impacted by the coupling, as is this presumably the main advantage of having such a coupled model.

We appreciate the reviewer's recognition of the value in our hybrid modeling approach. We agree that wildfire prediction, given its inherent unpredictability and sensitivity to numerous climatic, ecological, and anthropogenic factors, stands to benefit substantially from data-driven methods. In response to the reviewer's concerns, we have thoroughly revised our manuscript, specifically enhancing the model description section to clarify the underlying architecture and the integration between the ML fire model and the ELM land surface model. We have also expanded our validation section to address the reviewer's recommendations.

**Regarding the model description (Section 2):**  Though I appreciate that the underlying land-surface model (ELM) and XGBoost wildfire model have been documented previously (though, the current XGB implementation appears slightly updated re. the datasets used, time period etc.), given that the coupling between these models is the central development of this manuscript I felt that the details of the models (particularly XGB) and the coupling were a bit brief, and it was hard to figure out the answer to certain relevant details.  In particular:

- What are the respective model timesteps, and what is the coupling timestep?  The XGB model was trained (I think?) to predict GFED5 monthly BA, so does this mean it runs on a monthly timestep?  But, on P6, L62-63, the authors say "All the datasets are resampled to 0.25×0.25 spatial and annual temporal resolution" - so does this mean that it actually runs on an annual timestep?  But then, in the coupled model, it's described that the output of XGB is passed to ELM to affect land surface properties at the next timestep, and vice versa - I don't fully understand how this works if XGB is being trained with annual inputs to predict monthly or annual GFED5 BA.  ELM (I would assume?) has a much shorter timestep than annual/monthly, at least for properties like surface fluxes, soil moisture, LAI etc., as well as for the meteorological driving data used as input to the coupled model?

We apologize for the confusion. The ML fire model is trained at a monthly scale. While the ELM-BGC is driven by hourly meteorological forcing, the simulated surface meteorology and vegetation conditions are aggregated (averaged or accumulated) to a monthly scale. At the end of each month, these aggregated variables are fed into the ML fire model to predict burned areas, which are then used to update the vegetation properties. Section 2.3 has been rewritten for clarification. Please see the end of the response to this comment.

- What was the spatial resolution of ELM - does it match the 0.25 degree GFED5 grid that the XGB model (presumably?) provides output on?

Correct, both ELM-BGC and XGB file models have 0.25-degree resolution. We have clarified the model spatial resolutions in Section 2.3.

- There's insufficiently clear information about the XGB training process - the details are spread out in different parts of Section 2, and it's hard to work out exactly what were the inputs (including the time resolution, any dimension reduction that was applied, etc.), what was the target output, and what data was reserved for training vs validation.

We have rewritten Section 2.3 to clarify the input and output and the training process.

- "To reduce overfitting, we build a separate ML model for each year from 2001 to 2020 using the remaining 19 years' data" - I would stress that I'm not an ML expert, so maybe this is a simple lack of subject knowledge on my part.  But it's unclear to me what is meant here - how is a separate model trained for each year, using data from other years?  If the model is trained to predict the BA in one year based on the meteorological data of other years, it's not clear how it would learn the correct relationships.  Or do the authors mean, that it is trained to predict BA relationships for all the other years, and then the trained model is applied to the one year that was left out, as the validation data?  I'd appreciate if this could just be clarified a bit.  Additionally: does this mean that the model(s) can only be used for years between 2001-2020?  If so, that would seem to greatly limit it's usefulness for exploring future scenarios.

This is an excellent point, which was also raised by Reviewer #1. To address this, we have modified our training approach to build a canonical model. In the revision, we randomly split the 20-year monthly data into training and validation datasets, accounting for 80% and 20% of the entire dataset, respectively. The offline-XGB model is trained using only the training dataset to learn the relationship between the predictors and the burned area. Then, the offline-XGB is applied to the validation dataset to evaluate model performance on data not used during training.

- As with the points above - the nice schematic (Fig 1) shows the same meteorological and fire-specific input datasets being passed to ELM and the process-based fire model as to the XGB model, but it's unclear whether these inputs are provided at the same temporal and spatial resolution to the respective models, or whether there are intermediate pre-processing steps.  I'm not sure how easy this is to depict in the schematic, but as mentioned it's also a little unclear from the text and table of inputs as well.

Fire-specific inputs, such as lightning, GDP, and population density, are aggregated annually. ELM-BGC, which runs with hourly meteorological forcing data, retrieves this

information based on the year of the current timestep. The process-based fire model is called to simulate the burned area every hour, while the XGB model is trained and used at monthly intervals. To achieve this, we average the hourly meteorological conditions and simulated vegetation properties to obtain monthly means, and interpolate fire-specific variables from annual to monthly intervals using the nearest neighbor method. The XGB model is then trained at monthly intervals. When the offline-XGB model is coupled with ELM-BGC, the XGB model is called at the end of each month, while the ELM-BGC runs at hourly timestep. The hourly variables are accumulated internally to calculate the monthly mean at the end of each month.

We found it challenging to illustrate this in Figure 1; however, we have clarified it in the accompanying text.

- As I understand it, the XGB model is initially trained using PFT distributions diagnosed from a prior run of the ELM model using its processed-based fire scheme. However, the process-based scheme predicts a different fire distribution to the XGB model. Does this therefore introduce an inconsistency, i.e. that the XGB model is trained on PFT distributions that are predicated on the wrong spatial distribution of fires? Could this be improved by e.g. repeated iterations of running the ML4Fire-XGB coupled model to update the PFT distributions, and then re-training the XGB model? It would be good if the authors could comment on this.

This is another great point. The reviewer also suggested an approach that could enhance consistency between the XGB model and ELM-BGC. Currently, ELM is configured in the "biogeochemistry" (BGC) model, with PFT distributions prescribed based on satellite products. We have clarified this in the revised manuscript. Additionally, we are implementing our hybrid approach to ELM-FATES (Functionally Assembled Terrestrial Ecosystem Simulator), which will enable updates to account for the impacts of dynamic vegetation and plant demography.

Overall, regarding the comments about Section 2. We have rewritten this section as follows to address the comments.

*Initialized with the quasi-equilibrium state from the spin-up simulation, we conduct transient simulations with the process-based fire model in the ELM-BGC, driven by hourly NLDAS-2 meteorological forcings at a 0.25° resolution from 2001 to 2020. The process-based fire model operates on an hourly basis, matching the frequency of the meteorological inputs, while the ML fire model is trained and applied at a monthly interval, consistent with GFED5 data intervals. For training the offline-XGB model, the ELM-BGC outputs, including LAI, surface soil moisture, and PFT fractions, are averaged to monthly intervals, combined with monthly mean meteorological conditions, socioeconomic variables (GDP, population density), and lightning (as detailed in Table 1) to learn the relationship between predictors*

*and burned area. To reduce overfitting, the 20-year dataset is split, with 80% used for training and 20% for validation. During training, grid cells with fewer than 30 months of non-zero burned area (~two-thirds of the total number of grid cells) are masked. This step is important to avoid feeding the ML model distinct predictor combinations that all correspond to zero burned areas, which could skew the model's learning process. Model performance was evaluated based on its accuracy in predicting the spatial distribution and temporal variation of burned areas. Validation metrics included root mean square error (RMSE) and the coefficient of determination (R2).*

*We then integrate the offline-XGB to ELM-BGC, forming the coupled model ELM2.1-XGBfire1.0. The coupled model runs at 0.25° and hourly resolutions, where the hourly model predictions are accumulated to calculate monthly means. At the end of each month, the ML fire model is called to predict the monthly burned area, updating the land surface properties (LAI, PFT fraction, and soil moisture).*

**Regarding the comparison of burnt area results against four FireMIP models:**

- Why those particular 4 models?  E.g., the authors note that none of the models they compare against included a crop fire scheme, which is potentially one reason for poor performance over central US.  However a couple of the FireMIP models that are not included here, did have crop schemes - so it seems odd to omit these.  To be clear, I fully expect the ML4Fire-XBG model to outperform all the FireMIP models, it just seems a bit arbitrary why the comparison is made only against these four, out of nine FireMIP models that were included in the Rabin et al. (2017) paper.  If, for instance, these were the four best performing models over the CONUS, then it could make sense to compare against just these rather than against all of them.  But if that is the rationale, I couldn't see it mentioned anywhere (happy to be corrected though).

- All the figures comparing burnt area are described as an average over 2001-2020. However, the FireMIP experiments that are cited in Rabin et al. (2017) only went up to 2013.  Even the most recent round of FireMIP (aka ISIMIP3a) I think only goes up to 2019. So as far as I can understand, the FireMIP data can't be for the same time period.

We understand the reviewer's concern. In our manuscript, we used outputs from FireMIP Phase II (also known as ISIMIP3a), in which one of this paper's coauthors participated. Although there is no specific protocol paper for ISIMIP3a, two relevant publications have recently been made available (Burton et al., 2024; Park et al., 2024).

We selected ISIMIP3a data because the fire simulations are conducted with updated fire models and with an extended simulation period to 2019. In alignment with the ISIMIP3a simulation period, we have adjusted our comparison period to 2001–2019. At the time we conducted this research, only four models had data available. We have now updated our analysis to include seven FireMIP models, as used in the latest FireMIP benchmarking study (Burton et al., 2024).

Reference:

Burton, C., Lampe, S., Kelley, D. I., Thiery, W., Hantson, S., Christidis, N., Gudmundsson, L., Forrest, M., Burke, E., Chang, J., Huang, H., Ito, A., Kou-Giesbrecht, S., Lasslop, G., Li, W., Nieradzik, L., Li, F., Chen, Y., Randerson, J., Reyer, C. P. O., and Mengel, M.: Global burned area increasingly explained by climate change, Nat Clim Change, 10.1038/s41558-024-02140-w, 2024.

Park, C. Y., Takahashi, K., Fujimori, S., Jansakoo, T., Burton, C., Huang, H., Kou-Giesbrecht, S., Reyer, C. P. O., Mengel, M., Burke, E., Li, F., Hantson, S., Takakura, J., Lee, D. K., and Hasegawa, T.: Attributing human mortality from fire PM2.5 to climate change, Nat Clim Change, 10.1038/s41558-024-02149-1, 2024.

- The authors don't mention or discuss (as far as I could see) some very important caveats which really need to be attached to the comparison with FireMIP models. In particular, it should be noted that the process-based FireMIP models were run with different reanalysis driving data, at a much coarser spatial resolution. I appreciate that being able to run much higher resolution is a potential advantage of using an ML model. But, it needs to be acknowledged that it's not a like-for-like comparison of pure model process accuracy. The different driving data (from a different, global reanalysis product, provided at a much lower resolution than the North America-specific reanalysis data that the XGB model is driven by) is potentially a very important factor - a fairer comparison of performance would be to run the XGB model driven by the FireMIP driving data.

Thank you for pointing this out. In our model simulation, we adopted the same lightning, $CO_2$, population density, and GDP data used in ISIMIP3a, with the exception of the climate forcing data. To focus on fires in CONUS, we applied the upscaled hourly NLDAS-2 climate forcing at a spatial resolution of 0.25°, rather than the daily GSWP3-W5E5 forcing at 0.5° used in ISIMIP3a. This different reanalysis data source and differences in the spatial and temporal resolutions of the climate forcing could contribute to variations in burned area predictions.

Besides ISIMIP3a models, we also conducted ELM-BGC (with built-in process-based fire model) simulations driven by the same set of climate and socioeconomic forcing data as used to drive the coupled model ELM2.1-XGBfire1.0. The results show that the burned area simulation in ELM-BGC remains unsatisfactory, indicating that changes in climate forcing alone do not account for all limitations in burned area simulations in process-based models (at least for ELM-BGC). We have added the following discussion to the revised manuscript to clarify this point (Line 360-363).

*The ISIMIP3a models were driven by daily GSWP3-W5E5 forcings at a 0.5° spatial resolution. Differences in forcing data could lead to variations in burned area predictions. However, since both ELM-BGC and ELM2.1-XGBfire1.0 are driven by the same set of forcings, this suggests that limitations in physical understanding may significantly hinder the performance of the process-based model.*

Regarding the discussion and model validation:

- As mentioned above, it would be good to have some more quantitative discussion on the advantages of having the coupling, e.g. for vegetation distribution

Thank you for highlighting this point. First, we would like to clarify that in our current configuration, ELM is set up as a "biogeochemistry" (BGC) model, with PFT distributions prescribed based on satellite observations; we have added this clarification in the revised manuscript. Additionally, we are working to implement a hybrid approach within ELM-FATES (Functionally Assembled Terrestrial Ecosystem Simulator), which will allow the model to dynamically update vegetation and plant demographics, incorporating the impacts of changing vegetation structure over time.

Second, within the CONUS, fires primarily impact the terrestrial carbon cycle at localized scales, and their broader influence across the entire region is limited. ELM-BGC, like many terrestrial models, currently exhibits a significant bias in gross primary production (GPP) predictions across the CONUS (refer to the figure below). The coupled model, ELM2.1-XGBfire1.0, provides a significant improvement in fire prediction and slightly reduces the GPP underestimation compared to ELM-BGC, though this effect remains limited. Furthermore, converting burned area into carbon loss involves several uncertain parameters, which we did not optimize in this study.

Nonetheless, the coupling remains valuable, especially for higher-resolution model runs, examining fire-induced tree mortality, post-fire recovery, fire emissions, and fire-related air quality issues. This importance is amplified when ELM is run with its atmospheric component, E3SM, where the influence of fires on air quality, cloud, and surface meteorology becomes more significant.

[Figure]

Figure. Observed and simulated GPP (gC m-2 yr-1) averaged over the period 2001–2019. The dataset names are listed at the top of panels a and b. Panel c shows the difference between ELMv2.1-XGBfire1.0 and ELM-BGC. The numbers indicate the mean (M), bias (B), pattern correlation (R) against MODIS, and difference (Diff) between the two models in panel c. Black contours outline the eco-regions.

The following paragraph has been added to the discussion in our revision (Line 382-390).

*Although ELM2.1-XGBfire1.0 significantly improves the simulation of burned areas, its impact on terrestrial carbon fluxes remains limited. Within the CONUS, fires primarily affect the terrestrial carbon cycle at localized scales due to the relatively small burned areas. ELM-BGC, for instance, underestimates gross primary production (GPP) by approximately 30% (figure not shown). With more accurate fire predictions, ELM2.1-XGBfire1.0 helps to slightly reduce this negative bias (less than 1%). Additionally, while ELM-BGC using prescribed PFT distributions can moderate the effects of fires on the ecosystem, it does not account for fire-induced shifts in vegetation species, where species with greater resistance or fire-adaptive traits may gradually dominate.. Nonetheless, the coupling remains valuable, especially when the model is configured at higher resolutions. It is particularly important for evaluating fire-induced tree mortality, post-fire recovery, fire emissions, and their subsequent impacts on air quality, cloud formation, and surface meteorology, particularly when ELM is run as part of the E3SM.*

- All the comparison of BA performance is performed against GFED5, which is the same data that the model was trained on. Arguably, it's quite unsurprising that an ML model trained to predict GFED5 over CONUS from 2001-2020, would do better at predicting GFED5 over CONUS from 2001-2020, compared with global process-based models that were not specifically optimised to do this. Ideally, performance would be evaluated with out-of-sample tests - for example, by running the ML4Fire-XGB model with the FireMIP inputs as mentioned previously, or by comparing against alternative datasets and/or over different time periods to the training period.

We thank the reviewer for this valuable comment. We agree that the global process-based models were not specifically optimized for GFED5 and, importantly, not tailored for the CONUS region. Historically, the CONUS has not been a fire-prone area, and it has unique fire characteristics that may not be well represented in global models. Our analysis underscores that critical processes relevant to fire activity in CONUS may be missing in these global fire models. Improving the physical understanding of these processes and refining model parameters could enhance the performance of process-based models in capturing fire regimes over the CONUS.

In response to the reviewer's suggestion, we have added a new figure (Fig. 3, also shown below) to include GFED4s and FireCCI5.1 as additional reference datasets to account for observational uncertainties. Both GFED4s and FireCCI5.1 show a comparable spatial distribution to GFED5, with a spatial correlation coefficient exceeding 0.66. GFED5 includes more small fires, which are not detected in GFED4s and FireCCI5.1, leading to a burned area estimate that is 110% larger than the other two datasets. All process-based models overestimate burned areas when compared to GFED5, suggesting an even greater overestimation relative to GFED4s and FireCCI5.1.

[Figure]

*Figure 3: Observed burned area fraction (% yr-1). (a) GFED5 (2001-2019), (b) GFED4s (2001-2016), and (c) FireCCI5.1 (2001-2019). The numbers indicate the mean (M) burned area fraction and burned area (in Mha) in brackets for each dataset. The pattern correlation (R) against GFED5 is also shown, with an asterisk (*) denoting significance at the 0.01 level. Black contours outline the ecoregions.*

- On that note: the authors assume GFED5 is the ground truth in evaluating that the XGB model outperforms process-based models, but it should be acknowledged that there is a large uncertainty in the observation-based BA data. This observational uncertainty should also be addressed, for example by including comparisons not just against GFED5, but against alternative BA datasets that are available, for example the USGS Landsat Burned Area product for the CONUS, or one of the FireCCI BA products.

We have added GFED4s and FireCCI51 in the revision. Please see the response above.

- P16, L51-53: "However, the ML-fired process exhibits high accuracy, as demonstrated by the Offline-XGB model, making it a reliable tool for evaluating the fired area under different warming scenarios" - the authors show that the XGB model captures well the trend in GFED5 BA due to warming during the 2001-2020 period. However, this is the same period that the model was trained to perform well on. How reliable an indication is this that the model will still be accurate under high-end future warming scenarios, where the degree of climate change over the US will substantially exceed that observed over 2001-2020? This should be discussed.

We have removed the warming-temperature experiment from this paper, as suggested by Reviewer #1. We appreciate the reviewer's insight, and we are evaluating the model responses to high-end future warming scenarios in a separate study.

Minor/technical clarifications:

P2, L27-29: "Over the globe, climate change has contributed to a 16% increase in the global burned area over the past two decades, while human influences, including ignition and suppression, have reduced by 27% (Burton et al., 2023; Jones et al., 2022)" - as it reads, I don't think this isn't an accurate paraphrasing of the studies being referenced.

Currently it reads (to me) like: there has been a 16% increase in BA over the last two decades, to which climate has been a main driver, while the influence of humans has reduced by 27%. This isn't what either of these studies showed. Burton et al. find (based on FireMIP model data) that climate change since 1901 has made average BA (median over the 2003-2019 period) 16% higher than it would have been if climate stayed fixed at circa ~1901. However, they also find that human influences have made median BA 19% lower in the present compared with early 20th Century, suggesting the net effect over the 20th Century is a small decline in BA. Jones et al. show that, in MODIS BA data, total global BA has declined by 27% over the last two decades. This is similar to previously reported results from GFED4 and GFED5, which both show ~24% declines in total global BA over the last two decades. The reason for this decline has been attributed mainly to human influences (c.f. also Andela et al. 2017).

(As an aside: since the present manuscript was submitted, the Burton et al. (2023) pre-print that is referenced has now been published as a final article, and so the citation should be updated accordingly: https://www.nature.com/articles/s41558-024-02140-w)

Thank you for pointing this out. The abovementioned studies did show that direct human influences (suppression, agriculture expansion) have played a negative role in the increased global burned area. We apologize for the confusing words in the previous sentence. This clarification has been made in the revised manuscript (Line 27-31) with the paper citation updated.

*Globally, modeling studies show that climate change since the early 1900s has contributed to a 16% increase in the total burned area; however, human activities have led to a 19% decrease, resulting in a slight net decline in burned area over the 20th century (Burton et al. 2024). In the past two decades, satellite-derived data suggest that the global total burned area has declined by over 20%, with this trend primarily attributed to human influences (Jones et al. 2022; Andela et al. 2017).*

Section 2.2.3: From the looks of it, the authors use existing Level 1 EPA ecoregions for their analysis regions 1, 2, and 3, but then for their regions 4 and 5 they split the Eastern Temperature Forest Level 1 EPA region into two. What was the rationale for subdividing this region but not the others? Also, this section seems oddly located in the middle of the model description, in between the description of the individual model components (2.1) and the description of the coupling (2.3), even though it relates only to the analysis of the final results and is not part of the model description. It would be better to have this section on analysis methods after all the description of the models and model coupling, I think.

We used a combination of Level I and Level II EPA ecoregions. In the Level II data, Regions 4 and 5 are further divided: Region 4 (southeastern U.S.) includes the Southeastern Plains and Appalachian Forests, while Region 5 (northeastern U.S.) includes the Mixed Wood Plains and Atlantic Highlands. These two regions exhibit distinct fire patterns, as shown in Figure 3 of the manuscript.

Following your suggestion, we have moved this section to follow the descriptions of the models and model coupling. Thank you for the helpful feedback.

P8, L04-05: "According to the GFED5, the CONUS experiences an averaged burned area fraction (BAF) of 0.6–0.9% yr-1 (4.8–7.1 Mha yr-1), which is consistent with Chen et al., (2023)" - Not quite sure what the authors intended here. Chen et al. (2023) is itself the GFED5 burned area description paper, so trivially GFED5 burnt area is consistent with itself.

"Which is consistent with Chen et al. (2023)" has been removed from the revision.

P8, L05-06: "High-burned areas are predominantly observed in the WUS" - this seems an confusing statement, since the authors then go on to list other areas which have higher BA than the WUS, and indeed Figure 3 seems to show other areas of the US where BA is higher and more widespread.

This sentence has been rewritten as:

*The BAF over the WUS (Western Forested Mountains and NA Desert) ranges between $0.4$–$0.9\%$ $yr^{-1}$ ($1.1$–$2.3$ Mha $yr^{-1}$).*

P10, L33-34: "indicating that the ML model effectively describes crop fire thereby utilizing data on crop fraction and LAI" - Is this referring to the crop PFT fraction in the ELM model? (Rather than agricultural land use fraction, which isn't listed as an input)?

Yes. It refers to the crop PFT fraction in the ELM model.

P10, L52-53: "Notably, none of the process-based models has activated the explicit cropland fire model. That says all vegetation models treat pastures as natural grasslands." - This statement is slightly confusing and conflates two things. Pasture is not the same as cropland, and they are usually represented as different land cover types in DGVMs. Similarly crop residue burning is a very different fire regime to pasture fires.

Thank you for raising this point. We agree that there are fundamental differences between cropland fires and pasture fires. To clarify, all FireMIP models in this study exclude cropland fires (Burton et al., 2024; Extended Data Table 1). Additionally, all models except the LPJ-GUESS DGVMs do not explicitly represent pasture as a separate land cover type and, therefore, do not include pasture fires (Teckentrup et al., 2019). We have revised this sentence for clarity in Lines 276-279.

*As noted by Teckentrup et al. (2019) and Burton et al. (2024), none of the process-based models has activated the explicit cropland fire model. Fires are allowed in pastures. While LPJ-GUESS-SIMFIRE-BLAZE incorporates harvesting in pastures, reducing biomass and*

*influencing fire dynamics, all other process-based vegetation models treat pastures as natural grasslands for both vegetation growth and fire processes.*

---

## Author Response (AR2)

We sincerely appreciate the referees for their valuable and insightful comments on our manuscript. The feedback is instrumental in enhancing the quality and clarity of our research. This response document provides a detailed account of the changes implemented in relation to each specific comment from the referee. For ease of reference, referee comments are presented in black, while author responses are highlighted in blue, with modifications to the manuscript in italic font.

**Referee #1**

Thank you for your revisions in response to our reviewer comments. I feel that all mine have been satisfactorily addressed. I just have a few minor additional comments of a technical nature:

- The citation of Smith et al. (2001) for LPJ-GUESS-SIMFIRE at line 153 should be replaced with Knorr et al. (2016, Nature, https://www.nature.com/articles/nclimate2999).

Corrected. Thanks.

- L373: "employs the" should just be "employ"

Corrected.

- L383: "complex" should be "complicate"

Done.

- L383-386: Various corrections needed in this sentence. Suggested rewrite: "The fire-vegetation feedbacks further complicate this problem, with more complex dynamic vegetation models being slow to reach equilibrium after disturbances. The choice of prescribed or dynamic vegetation could also play a role; note that among all the process-based models, CLASSIC, VISIT, and ELM used prescribed vegetation while all others used dynamic vegetation."

Corrected. Thank you for the suggestion.

- L387: "institute" should be "institutes".

Corrected.

**Referee #2**

I greatly appreciate the authors' efforts in responding comprehensively to my comments. I have a couple of very minor additional clarifications below, which relate to parts of new text added to the manuscript, but otherwise am happy to recommend the paper be accepted for publication in GMD.

Additional minor/technical comments (line numbers refer to clean version of revised manuscript):

L26-28 "In the past two decades, satellite-derived data suggest that the global total burned area has declined by over 20%, primarily attributed to human influences. The continental United States has emerged as a hotspot for wildfires..." - These statements individually are absolutely fine, but possibly now reads a little oddly as the second sentence no longer really follows on from the preceding sentence. The authors are very welcome to keep it as is, but they could consider changing the following sentence to "*However*, the continental United States has emerged as a hotspot for wildfires..." to make for (in my opinion) a slightly more punchy introduction. Entirely at the authors' discretion though.

Thanks for the suggestion. We have added a "However" before the second sentence for a better flow.

L181-183 "Lightning, population density, and GDP data are resampled to 0.25°×0.25° spatial using bilinear interpolation and annual temporal resolution using the nearest neighbor method" - I want to double check that this revised description is correct, specifically re. the lightning dataset. The authors state that they use the NASA LIS/OTD 2-hourly lightning climatology, and so the idea of using nearest neighbour interpolation to resample this to an annual value, before then re-interpolating it to a monthly value (which is what the ML fire model uses as input), seems slightly odd. A nearest neighbour interpolation of 2-hourly lightning data would correspond to picking a single 2-hour time period and using that value for the whole year, which seems unnecessary when the 2-hourly data could just be used to directly calculate a monthly-mean climatology to drive the model with. It also seems at odds with the authors' description in the response to reviewers document, where they indicate that nearest neighbour interpolation was used to interpolate values which were already at annual resolution, like the GDP and population density, to monthly values to use as inputs to the model - which makes more sense. So, I just wanted to check whether this sentence in the revised manuscript is correct.

Thank you for pointing out this important distinction; this is indeed a valuable clarification. The lightning data are treated differently from the population density and GDP datasets. Specifically, the population density and GDP datasets are provided annually, and since their monthly variations are considered less relevant for fire prediction, each annual value is simply assigned uniformly across all months within the corresponding year. In contrast, the lightning data, originally available as a 2-hourly climatology from NASA LIS/OTD v2.2, are first aggregated by summing the 2-hourly data into monthly climatological means. These monthly climatologies remain constant and are repeated uniformly across all years of the simulation period, disregarding interannual variations. We have revised the manuscript text as follows to accurately reflect this approach:

*The 2-hourly climatology lightning flashes data from NASA LIS/OTD v2.2 at 2.5° resolution are used to calculate the number of natural ignitions. Lightning data are aggregated by summing the 2-hourly data to derive monthly climatological means, and these monthly climatologies are repeated across all years, disregarding interannual variations. The annual gridded population density data is acquired from Goldewijk et al. (2017), while the GDP per capita is from the World Bank (https://data.worldbank.org/), which are assigned constant values for all months within each corresponding year. All datasets are spatially resampled to a 0.25°×0.25° grid using bilinear interpolation.*